# An assessment of prevalence of Type 1 *CFI* rare variants in European AMD, and why lack of broader genetic data hinders development of new treatments and healthcare access

Amy V. Jones[1], Darin Curtiss[1], Claire Harris[1], Tom Southerington[2,3], Marco Hautalahti[2], Pauli Wihuri[2], Johanna Mäkelä[2], Roosa E. Kallionpää[4], Enni Makkonen[5], Theresa Knopp[6], Arto Mannermaa[7], Erna Mäkinen[8], Anne-Mari Moilanen[9], Tongalp H. Tezel[10], on behalf of the SCOPE Study group[¶], Nadia K. Waheed [1,11]*

1 Gyroscope Therapeutics Limited, London, United Kingdom, 2 Finnish Biobank Cooperative–FINBB, Turku, Finland, 3 University of Turku, Turku, Finland, 4 Auria Biobank, Turku University Hospital and University of Turku, Turku, Finland, 5 Finnish Clinical Biobank Tampere, Tampere, Finland, 6 Helsinki Biobank, HUS, Helsinki University Hospital, Helsinki, Finland, 7 Biobank of Eastern Finland, KYS, Kuopio, Finland, 8 Biobank of Central Finland, Hospital Nova of Central Finland, Jyväskylä, Finland, 9 Biobank Borealis of Northern Finland, Oulu University Hospital, Oulu, Finland, 10 Department of Ophthalmology, Edward S. Harkness Eye Institute, Columbia University College of Physicians and Surgeons, Columbia University Medical Center, New York, NY, United States of America, 11 Department of Ophthalmology, Tufts University School of Medicine, Boston, Massachusetts, United States of America

¶ Membership of the SCOPE Study group is provided in the Acknowledgments.
* nadiakwaheed@gmail.com

## Abstract

### Purpose

Advanced age-related macular degeneration (AAMD) risk is associated with rare complement Factor I (FI) genetic variants associated with low FI protein levels (termed 'Type 1'), but it is unclear how variant prevalences differ between AMD patients from different ethnicities.

### Methods

Collective prevalence of Type 1 *CFI* rare variant genotypes were examined in four European AAMD datasets. Collective minor allele frequencies (MAFs) were sourced from the natural history study SCOPE, the UK Biobank, the International AMD Genomics Consortium (IAMDGC), and the Finnish Biobank Cooperative (FINBB), and compared to paired control MAFs or background population prevalence rates from the Genome Aggregation Database (gnomAD). Due to a lack of available genetic data in non-European AAMD, power calculations were undertaken to estimate the AAMD population sizes required to identify statistically significant association between Type 1 *CFI* rare variants and disease risk in different ethnicities, using gnomAD populations as controls.

### Results

Type 1 *CFI* rare variants were enriched in all European AAMD cohorts, with odds ratios (ORs) ranging between 3.1 and 7.8, and a greater enrichment was observed in dry AMD

**Data Availability Statement:** All relevant data are within the paper and its Supporting Information files.

**Funding:** This study was funded by Gyroscope Therapeutics Limited. Study design, data collection, data analyses, decision to publish and preparation of the manuscript was performed by employees of Gyroscope Therapeutics Limited (https://www.gyroscopetx.com/).

**Competing interests:** The authors have declared that no competing interests exist: Tom Southerington Enni Makkonen Roosa E. Kallionpää Pauli Wihuri Theresa Knopp Marco Hautalahti Johanna Mäkelä Arto Mannermaa Erna Mäkinen Anne-Mari Moilanen Authors with competing interests I have read the journal's policy and the authors of this manuscript have the following competing interests: Amy Jones: employee and share holder of Gyroscope Therapeutics Ltd Darin Curtiss: employee and share holder of Gyroscope Therapeutics Ltd Claire Harris: employee and share holder of Gyroscope Therapeutics Ltd, Research grant from RA Phaemaceutics (payment to institution), Royalty income from commercialized factor I ELISA; Hycult Biotech, consultancy income from Q32 Bio Inc, Chinook Therapeutics, and Biocryst Pharmaceuticals (all payment to institution), Nadia Waheed: employee and share holder of Gyroscope Therapeutics Ltd, grants from Carl Zeiss Meditec, Topcon, Regeneron, Heidelberg, Nidek, Optovue, consultancy income from Apellis, Nidek, Boehringer Ingelheim, stock in Ocudyne.

**Abbreviations:** AAV, Adeno-Associated Virus; ACMG, American College of Medical Genetics and Genomics; AMD, Age-related macular degeneration; AAMD, Advanced Age-related macular degeneration; AP, Alternative pathway; CI, Confidence interval; FI, Factor I; FH, Factor H; CNV, Choroidal neovascularisation; CS, Complement system; FINBB, Finnish Biobank Cooperative; GA, Geographic atrophy; GnomAD, Genome Aggregation Database; GWAS, Genome-wide association study; IAMDGC, International AMD Genomics Consortium; MAF, Minor allele frequency; MNV, Macular neovascularisation; NGS, Next-generation sequencing; OR, Odds ratio; RAF, Rare allele frequency; RPE, Retinal pigment epithelium; PheWAS, Phenome-wide association studies; PCA, Principal component analysis; SNP, Single nucleotide polymorphism; USA, United States of America; UTR, Untranslated; UKB, UK Biobank; VAF, Variant allele frequency; VEGF, Vascular endothelial growth factor; WGS, Whole exome sequencing.

from FINBB (OR 8.9, 95% CI 1.49–53.31). The lack of available non-European AAMD datasets prevented us exploring this relationship more globally, however a statistical association may be detectable by future sequencing studies that sample approximately 2,000 AAMD individuals from Ashkenazi Jewish and Latino/Admixed American ethnicities.

## Conclusions

The relationship between Type 1 *CFI* rare variants increasing odds of AAMD are well established in Europeans, however the lack of broader genetic data in AAMD has adverse implications for clinical development and future commercialisation strategies of targeted FI therapies in AAMD. These findings emphasise the importance of generating more diverse genetic data in AAMD to improve equity of access to new treatments and address the bias in health care.

## Introduction

Age-related macular degeneration (AMD) is a progressive retinal disease that is the leading cause of irreversible central vision loss among the elderly population across industrialised countries [1–3]. Early stages of AMD are characterized by drusen and pigmentary changes, causing mild visual impairment. Many patients progress to late stage or advanced AMD (AAMD), which can be distinguished into two subtypes; wet AMD involves angiogenesis in the choroid and subsequent macular or choroidal neovascularisation (MNV or CNV), and advanced dry AMD involves degeneration or geographic atrophy (GA) or degeneration of the retinal pigment epithelium (RPE) and overlying retina [4]. The overall prevalence of AAMD in the US population 40 years and older has been estimated to be 1.47%, with disease rates increasing significantly with age [1]. AMD is a complex disease with a multifactorial aetiology influenced by age and environmental factors such as diet and sunlight exposure, and a positive family history [3, 5]. Underlying ethnicity is known to influence disease prevalence [1], clinical presentation [6], and treatment response [7], but a lack of studies performed in patients from non-European backgrounds makes understanding the breadth and impact of this challenging. Unlike wet AMD, to date there is no effective treatment for dry AMD, and while anti-vascular endothelial growth factor (VEGF) therapy has been reasonably successful in the short term, vision gains are not maintained at 5 years after starting treatment, and many patients ultimately progress to GA [8].

An established approach for developing effective therapies for diseases is to target underlying pathogenomic pathways, identified using results from human genetic studies conducted on affected patients [9]. One example is HMG-CoA reductase encoded by the *HMGCR* gene, that has been associated with serum cholesterol levels and is the target for statins [10]. Genetic evidence combined with results from immunohistochemistry and biomarker studies strongly point to an overactive complement system (CS) being a main driver of AMD development and progression, and this may be a suitable target pathway for targeted therapy [11, 12]. The CS is a complex and tightly regulated pathway critical to maintaining an effective innate immune response [13], and it remains to be determined what is the optimal point to intervene [14]. Injection of complement inhibitors into the ocular fluid has been one approach trialled in AMD with varying success [15–18]. Another strategy is using Adeno-Associated Virus (AAV) gene therapy to target human complement Factor I (*CFI*) gene to the retinal pigment

epithelium to drive expression of FI protein. FI is a serine protease that acts a global down regulator of all three CS pathways, acting in conjunction with co-factors, one of which is complement Factor H (FH) [19–21].

Ocular FI supplementation therapy may show the greatest treatment response in GA patients who carry particular rare genetic variants in the *CFI* gene, which causes them to express lower levels of FI protein in the body and in the eye [22–26]. The collection of rare *CFI* variant genotypes that cause low or 'haploinsufficient' FI protein levels are termed 'Type 1' [22, 26]. Type 1 rare *CFI* variants are amongst the strongest risk factors for AAMD [22, 23, 27, 28], with some genotypes linked to an earlier age of disease onset and altered disease progression [24, 29]. Genetically driven, sustained reduction in FI level over a person's lifetime may be considered causal for AAMD [30]. Consistent secretion of endogenous FI to physiological levels with AAV should balance the hyperactive CS, reducing chronic inflammation and ultimately slow macular degeneration [31]. A treatment response from FI supplementation is also anticipated in the broader GA population, as studies have shown complement dysregulation is widespread across individuals who carry a range of other rare and common risk-associated variants in complement genes [32, 33]. Clinical trials testing response to ocular FI supplementation delivered by AAV in GA are underway in phase 1 (NCT03846193) and two phase 2 studies (NCT04437368 and NCT04566445).

Grouping individuals based on particular disease-risk genotypes is a useful approach to stratify patients in the clinic and to test treatments targeting shared underlying pathological processes in a homogeneous population, like those that carry *CFI* rare variants. Estimations show Type 1 *CFI* rare variants are found in between 1–3% of GA patients from European backgrounds, but there is lack of concordance between which genotypes are defined 'Type 1' [25, 26, 34, 35]. Population genetic studies report that typically, rare genetic variants that confer a strong functional consequence tend to arise and cluster in specific populations or ethnicities, compared to more common genetic variants that are typically distributed across most human populations [36, 37]. It is unclear how relevant the prevalence estimations from Type 1 *CFI* rare variants derived from European AMD are for AMD patients arising from different geographies or ethnicities. Understanding this has implications for clinical development strategy and commercial planning for FI targeted therapies, where prevalence estimates may help inform who is most likely to benefit globally. In addition, these knowledge gaps have implications for equity of access to new treatments and health care.

To explore this in more detail, analyses of available evidence from different AAMD datasets was undertaken to determine whether Type 1 *CFI* rare variant genotype prevalences vary between cohorts. Frequencies observed in AAMD datasets from European and Finnish populations were compared to background prevalence rates sourced from publicly available data from the Genome Aggregation Database (gnomAD) [38]. Due to a lack of non-European AAMD cohorts with comprehensive targeted sequencing data available, an investigation was conducted to determine whether it was possible to use underlying background Type 1 *CFI* rare variant genetic frequencies of a population to predict the prevalence and enrichment of Type 1 *CFI* rare variants in those that might go on to develop AMD using data from gnomAD. This may stimulate the field to generate more sequencing studies in individuals with AAMD from diverse ethnic backgrounds, further supporting the clinical development of novel therapies which could reach a wider range of AAMD patients.

## Aims

To explore the prevalence of Type 1 *CFI* rare variants in available AAMD datasets and background populations from different ethnicities.

## Methods

### Overview of methods and selection of datasets

For *CFI* variant frequency analysis, we selected the 18 'Type 1' *CFI* rare variant genotypes defined by Java et. al. (2020) [26], in a European AAMD cohort, on the basis 29 carrier patient serum blood samples underwent comprehensive serum-based *in vitro* functional assays measuring FI level and complement activity to determine 'Type 1' status.

Because Java et. al. (2020) described the functional nature of different *CFI* rare variant from only in a minority of cases with serum available for functional evaluation of each genotype, more accurate variant frequency data was sourced from an earlier study examining the same underlying European AAMD (n = 2,266) and control sample (n = 1,400) datasets, as reported by Kavanagh et. al. (2015) [22], which employed a targetted next-generation sequencing strategy to capture the *CFI* coding and 5' untranslated (UTR) and 3' UTR regions (Table 1) [22].

Allele frequencies for rare *CFI* variants were compared to those observed in published studies providing genetic data from other unique AMD cohorts comprising greater than 500 individuals, and cohorts that did not filter their sample size based on other criteria prior to genotyping. The International AMD Genomics Consortium (IAMDGC) AAMD meta-GWAS publication [34] was selected as the source of variant frequencies for contributing studies published elsewhere [24, 39, 40], on the basis that duplicate individuals were filtered out during data analysis. Type 1 *CFI* rare variant status was determined in a series of AAMD and population control studies listed below. All Type 1 *CFI* rare variants identified in AAMD cases were recorded as heterozygous. Variant positions were provided to genome build hg19/GRCh37 coordinates and FI amino acid changes were described according to protein accession number NP_000195.2. The term 'rare' refers throughout to variants that are equal or less than 1% minor allele frequency (MAF) in European background (control) populations [41].

### Genetic datasets

**IAMDGC meta-GWAS.** The IAMDGC represents the largest collection of AMD samples (16,144 European AAMD cases and 17,832 European controls), and a detailed description of the study design and genetic data has been previously described [34]. To summarise, AMD samples were gathered from 26 studies from cohorts across the United States (US), Europe and Australia. AAMD cases were defined using a series of grading systems as presence of CNV

**Table 1. Type 1 *CFI* rare variants are collectively enriched in European AAMD datasets.**

| Study | AAMD cases (n) | Type 1 *CFI* rare variant carriers in cases (n) | Collective AAMD Type 1 *CFI* rare variant frequency | Controls (n) | CFI Type 1 rare variant carriers in controls (n) | Collective control Type 1 *CFI* rare variant frequency | P value | OR | 95% CI |
|---|---|---|---|---|---|---|---|---|---|
| SCOPE | 3,243 | 84 | 2.590% | 64,603 | 221 | 0.342% | 8.01E-78[a] | 7.75 | 6.02–9.99 |
| UK Biobank pheWAS portal [44] | 3,770 | 38 | 1.008% | 234,829 | 764 | 0.325% | 6.77E-13[a] | 3.12 | 2.25–4.33 |
| IAMDGC [34] | 16,144 | 197 | 1.220% | 17,832 | 53 | 0.297% | 2.74E-23[a] | 4.14 | 3.06–5.62 |
| FINBB | 943 | 2 | 0.212% | 12,562 | 3 | 0.024% | 4.20E-02[b] | 8.90 | 1.49–53.31 |
| Kavanagh et. al. (2015) [22] | 2,266 | 54 | 2.383% | 1,400 | 3 | 0.214% | 9.32E-09[b] | 11.37 | 3.55–36.43 |

CI confidence interval. Statistical significance was calculated using Chi Squared test[a] or Fishers Exact test[b]. n; number, OR; odds ratio, CI; confidence interval.

and/or GA in at least one eye as determined by the study ophthalmologist or principal investigator, and age at first diagnosis >50 years. The breakdown of GA, CNV or mixed AAMD subtype in European individuals was approximately 20%, 67%, and 13%, respectively. Controls were individuals without known advanced or intermediate AMD. Participant DNA was genotyped using a customised chip that tested both common variants and rare variants, including many rare protein-altering variants previously identified using targeted sequencing in a European AAMD cohort [42]. Participant data was filtered to exclude data from related or duplicate individuals, and principal component analysis was applied to define and group individuals by ethnicity, prior to GWAS analysis. AAMD individuals with non-European ancestries were also identified, however as each population totalled under 500, they were discounted from our investigation due to insufficient power for rare variant detection.

**UK Biobank.** The UK Biobank is a large-scale prospective cohort study that has so far enrolled 502,507 participants aged 40–69 years across the UK between 2006–2010 [43]. Exome sequencing data from 269,171 UK Biobank participants of European ancestry was recently combined with survey responses on sociodemographic characteristics, lifestyle, body measurements and medical records, that were mapped as binary or quantitative phenotypes in the form of ICD10-based codes, then subject to a series of phenome-wide association studies (pheWAS) [44]. Participant self-reported AMD status was mapped as a binary trait to the closest ICD10 phenotype category for AMD; 'Union#H353#H35.3 Degeneration of macula and posterior pole'. Summary genotype counts of affected cases and controls were obtained from the AstraZeneca PheWAS Portal (https://azphewas.com/) and evaluated for Type 1 *CFI* rare variant status.

**SCOPE.** GA patients were recruited into SCOPE, a natural history study (study number NCT03894020; https://clinicaltrials.gov/). Patients were included who had unilateral or bilateral GA as determined by FAF and a reading performance of ≥40 letters by best corrected visual acuity. Patients were excluded who had MNV or diabetic retinopathy, as determined by the principal investigator at the study site. Patients defined their race or ethnicity as either White, Hispanic or Latino, American Indian or Alaskan Native, Asian, Black or African American, Native Hawaiian or Other Pacific Islander, other, unknown, or not reported. Approximately 98% self-identified as White.

Patient's saliva DNA was screened for rare *CFI* variants using targeted next-generation sequencing conducted by Molecular Vision Laboratory (Oregon, USA). A customised capture panel was designed using Agilent SureSelect Target Enrichment kit to amplify the *CFI* coding region. Paired-end reads were sequenced using Illumina Miseq V2 platform, using acceptance thresholds of >30X coverage over >98% target region. Sequencing reads were aligned to human reference genome NCBI build GRCh37v3 using NextGENe software v2.4.2.3 and genotypes were called using Genetics Assistant v1.4.7 program.

**Finnish AAMD.** The Finnish Biobank Cooperative (FINBB) is a network of six hospital and two cohort university biobanks which provides a centralised resource for conducting genotype-phenotype studies (www.finbb.fi). We identified 943 individuals in FINBB with AAMD using ICD-10 coded biobank health-registry information; defined as a positive diagnosis of atrophic 'dry' AMD (H35.30), but excluding exudative 'wet' AMD and glaucoma (H35.31, H36.0, H35.7, H40.1, H40.2, H40.3, H40.4, H40.5, H40.6, H40.8, H40.9). Individuals were aged 55–95 years, and currently alive. Sequencing over the *CFI* coding region and 50 base pairs either side of intron/exon boundary was performed on de-identified DNA samples using a CleanPlex amplicon sequencing approach (Paragon Genomics) by the Finnish Institute of Molecular Medicine laboratory. PCR amplicons were sequenced as 151 base pair (bp) or 251 bp paired end reads and two 8 bp index reads using the Illumina MiSeq instrument (Illumina, San Diego, CA, US). The data were analysed with a novel in-house bioinformatics pipeline,

where sequencing reads were aligned using bowtie 2 (version 2.2.9) against human reference genome Ensembl (version 75) build of GRCh37. All *CFI* variants with variant allele count over 5 and variant allele frequency (VAF) over 0.5% were taken into consideration. From these candidates, false positives were initially filtered out based on the noise (estimated error rate) level from control sample in every run making use of a binomial distribution to compute p-value for the event that more than the frequency of alternative alleles were observed when the null hypothesis is true. However, variants with VAF over 2% were called independent of the noise. A specific frequency ratio was used to filter out false positive by dividing the ratio of variant calls/number of all the bases (at a position) by the ratio of variant allele quality sum/quality sum of all the bases. All variants were verified with Integrative Genomics Viewer [45].

**GnomAD.** The gnomAD database is a resource developed by international coalition of investigators to aggregate exome and whole genome sequencing data from a wide range of large-scale sequencing projects and make summary data available to the wider scientific community [38]. The database totals data from 141,456 unrelated individuals spanning 6 global and 8 sub-continental ancestries, from disease-specific or population genetics studies. Principal component analysis (PCA) and a random forest classifier was used to assign individuals to known ancestries (either African/African American, Latino/Admixed American, East Asian, South Asian, Non-Finnish European, Finnish, and other or population not assigned). Data from GnomAD release V2.1.1 was used for variant analyses for each ethnicity, apart from the African/African-American population where only V3.1.1 release was used, as this provided results from a greater number of individuals. V3.1.1 also provided data from Amish and Middle Eastern ancestries for the first time however due to small sample size (n<500) and absence of Type 1 *CFI* rare variants, they were not included in our analyses.

Data was accessed through the gnomAD browser (http://gnomad.broadinstitute.org) and we used 'observed Afs' representing the count ratio of the actually detected minor alleles to reliably sequenced alleles.

## Ethics statement

For the SCOPE and Finnish studies, written consent was obtained from all participants in accordance with the Declaration of Helsinki. For the SCOPE study, institutional review board (IRB)/ ethics committee/ regulatory authority approval was obtained on a local or country-specific basis; United Kingdom (Medicines and Healthcare products Regulatory Agency, London, and North of Scotland Research Ethics Committee), United States of America (Oregon Health & Science University, Portland, The Johns Hopkins Medicine IRB, Baltimore, Wills Eye Hospital IRB, Philadelphia, Columbia University IRB, New York and Advarra), Australia (Therapeutic Goods Administration, Woden, Bellberry Human Research Ethics Committee, Eastwood), Poland (Komisja Bioetyczna przy Bydgoskiej Izbie Lekarskej, Komisja Bioetyczna przy Okregowej Izbie Lekarskiej w Lodzi, Komisja Bioetyczna Slaskiej Izby Lekarskiej w Katowicach, Office for Registration of Medicinal Products, Medical Devices and Biocidal Products, Warszawa), Germany (Ethikkommission der Medizinischen Fakultät der Eberhard Karls Universität und am Universitätsklinikum Tübingen), France (Comité de protection des Personnes Sud Méditerrannée I–Marseille), The Netherlands (CMO Regio Arnhem-Nijmegen UMC St. Radboud) and Spain (CEIC-Hospital Clinico San Carlos, Madrid). For the Finnish study, written consent was obtained from all participants also in accordance with the Finnish Biobank Act (688/2012), and ethics review and approval was obtained from the Helsinki Ethical Review Board. Ethics review committees/IRBs that reviewed and approved previously published datasets who's summary findings were evaluated in this manuscript have been previously described; IAMDGC [34], UK Biobank [44], and gnomAD [38].

## Statistical analyses

The collective OR for Type 1 *CFI* rare variants in each population was calculated by totalling the number of carriers of each variant observed per dataset and assuming that all instances of *CFI* rare variant genotype identified were mutually exclusive and there was uniform sequencing coverage for all individuals within each sub-population. For each genotype, the MAF was calculated as the number of rare variant alleles divided by total number of alleles observed in the respective datasets, and the collective Type 1 *CFI* rare variant MAF was the sum of all Type 1 *CFI* rare variant genotype MAF values.

The association of collective Type 1 *CFI* rare variants with AAMD was calculated using Chi squared test, apart from those cohorts reporting frequency of <5, where we used Fisher's exact test. Sample size estimates and power calculations were performed using PROC POWER procedure in SAS (ver 9.4).

Variant American College of Medical Genetics and Genomics (ACMG) classifications [46] were obtained from Franklin, a web-based variant annotation tool (https://franklin.genoox. com) accessed on 19/01/2022.

## Results

### Evaluating Type 1 *CFI* rare variant prevalences in European AAMD datasets

For the 18 rare *CFI* variant genotypes categorised as Type 1 [26], collective frequencies from the Kavanagh et.al. (2015) AAMD dataset [22] were compared to total frequencies derived from four independent European AMD datasets. These studies include those with available paired case-control genotype data from the IAMDGC AAMD meta-GWAS [34] and a recent UK Biobank pheWAS data release [44]. We also sourced genotype data from individuals with GA from the natural history study SCOPE, and individuals with dry AMD from FINBB, a hospital and academic network of biobanks based in Finland. A description of key differences between study design is provided in S1 Table.

These analyses showed Type 1 *CFI* rare variants were significantly enriched in all European AAMD cohorts, with the strongest odds observed in the Kavanagh et. al. (2015) dataset [22] (OR 11.37, 95% confidence interval (CI) 3.55–36.43) (Table 1). Compared to the collective ORs from SCOPE, UK Biobank, and IAMDGC which ranged between 3.1 and 7.8, greater enrichment was observed in the FINBB AAMD dataset (OR 8.9, 95% CI 1.49–53.31). This observation is striking given the starting collective frequency of Type 1 *CFI* rare variants observed in both the FINBB AAMD and Finnish background control populations were approximately 5–15 fold lower compared to the other European AAMD or control populations.

Considering each Type 1 *CFI* rare variant genotype identified in the European AAMD dataset separately, in the Kavanagh et. al. (2015) AAMD dataset [22] five were observed with MAFs between 0.1%-1% (p.Ala240Gly, p.Gly119Arg, p.Gly162Asp, p.Ala258Thr and p. Gly287Arg), and the remaining 13 genotypes had MAFs under 0.1% (S2 Table). Similar individual MAFs were observed in the other non-Finnish European AMD datasets for four of the genotypes (p.Gly119Arg, p.Ala258Thr, p.Gly287Arg and p.Val152Met), but those with rarer MAFs (<0.1%) went largely undetected in the other AMD datasets, likely due to limited power for identification due to sample size limitations. Enrichment for Type 1 *CFI* rare variants observed in Finnish AMD was driven by the presence of only two *CFI* rare variant genotypes (p.Gly119Arg and p.Arg474Ter), despite their apparent absence in the gnomAD Finnish background control population (S2 Table).

## Evaluating Type 1 *CFI* rare variant prevalences in different background ethnicities in gnomAD

GnomAD is a publicly available resource providing precise genetic variant frequency data from different ethnic populations [38]. GnomAD allele frequency information may be considered representative of the expected allele frequencies in the given general (background) population, and these can be used as controls for comparison to disease allele frequencies in various genetic association studies, when paired controls may be unavailable [47]. Inspection of the gnomAD database showed that overall, collective Type 1 *CFI* rare variant allele frequencies and diversity varied considerably across the different gnomAD background ancestries (S3 Table). Interestingly, the highest background frequency was observed in the Ashkenazi Jewish population, who reported a 3 fold higher collective frequency (0.550%), compared to the background non-Finnish European population frequency (0.174%). The Latino/Admixed American population reported a similar collective frequency of 0.198% to the non-Finnish European population.

The lowest collective Type 1 *CFI* rare variant allele frequencies were reported in African/African-American (0.019%), Finnish (0.012%), South Asian (0.052%), and East Asian background populations (0.005%). For these ancestries, we currently lack sufficiently sized datasets of genetically defined AAMD and paired control datasets to determine whether our list of Type 1 *CFI* rare variants that were identified in a European AAMD cohort, are also significantly enriched in the AAMD populations that would arise from these ethnic backgrounds. We explored whether it was possible to determine what sample sizes of non-European AAMD might be required to try to answer this question, using gnomAD ethnic populations as controls.

The highest diversity of different genotypes from the Type 1 *CFI* rare variant list were observed in the background non-Finnish European population (72%; 13/18) (S4 Table). This was followed by African/African-American and Latino/Admixed American populations (both 33%; 6/18), South Asians (22%; 4/18), then the Finnish and Ashkenazi Jewish populations (both 11%; 2/18), and finally East Asians with 6% (1/18).

Assuming that collectively Type 1 *CFI* rare variants increase odds of developing AAMD of at least 3 fold and with available information in each reference database (the number of subjects and the rate of Type 1 *CFI* rare variants) [1], we then calculated the AAMD population size required to provide statistically significant association (P<0.05) with 80% power or 90% power, respectively. A series of power calculations were performed to predict the AAMD sample sizes that would be required for screening, using the allele frequency of Type 1 *CFI* rare variants observed in the gnomAD background ethnicities (Table 2).

At 80% power, a sample size of more than 1,800 would be required to detect significance from Type 1 *CFI* rare variant frequencies in a non-Finnish European AAMD cohort, which is supported by positive findings in the sample sizes examined in the SCOPE, UK Biobank and IAMDGC AAMD datasets (Table 1).

For the Latino/Admixed American and South Asian AAMD ancestries, sample sizes of 1,800 and 15,000 respectively would be required to detect significance. Given the higher Type 1 *CFI* rare variant frequency observed in the background Ashkenazi Jewish population, a smaller AAMD sample size of 670 would be needed to detect significant association of Type 1 *CFI* rare variants and AAMD.

The African/African-American, South Asian and East Asian background ethnicities all report lower collective Type 1 *CFI* rare variant background allele frequencies and smaller gnomAD control sizes compared to Europeans. Even if the AAMD sample sizes from these ethnicities were at 1,000,000, the power calculations predict there still would not be sufficient power

**Table 2. Projected AAMD sample sizes required to test for significant association between Type 1 *CFI* rare variants and disease in different ethnicities defined by gnomAD.**

| Ethnicity | GnomAD release | GnomAD sample size (n) | Collective Type 1 *CFI* rare variant frequency (%) | Projected AAMD sample size | |
|---|---|---|---|---|---|
| | | | | 80% power | 90% power |
| Ashkenazi Jewish | V2.1.1 | 5,185 | 0.550% | 670 | 1060 |
| Latino/Admixed American | V2.1.1 | 17,720 | 0.198% | 1,800 | 2,800 |
| Non-Finnish European | V2.1.1 | 64,562 | 0.174% | 1,800 | 2,700 |
| South Asian | V2.1.1 | 15,308 | 0.052% | 15,000 | 32,000 |
| African/African-American | V3.1.1 | 20,744 | 0.019% | * | * |
| Finnish | V2.1.1 | 12,562 | 0.012% | * | * |
| East Asian | V2.1.1 | 9,977 | 0.005% | * | * |

For each ethnicity, AAMD sample sizes were calculated using background control allele frequency, assuming OR = 3, and a significance threshold of $p < 0.05$, at 80% and 90% power.

*indicates statistical significance cannot be achieved even with an AAMD samples size of 1,000,000.

to detect a significant association between Type 1 *CFI* rare variants and AAMD arising in these populations.

The Finnish European population was also similarly low in starting background collective Type 1 *CFI* rare variant allele frequency (0.12%), and as illustrated in Table 2, the predicted AAMD population size needed to detect a significant association would have to be >1,000,000, before becoming informative. However direct sequencing of *CFI* in a Finnish AAMD cohort of only 943 participants revealed that collectively, Type 1 *CFI* rare variants doubled the odds of disease development than that observed in non-Finnish AAMD Europeans (OR 8.9 vs OR 3–6, respectively; Table 1).

## Discussion

### Are Type 1 *CFI* rare variants found only in European AAMD?

In this study, prevalences of Type 1 *CFI* rare variants previously defined by Java et. al (2020) [26] were reviewed in different datasets of European AAMD and compared to allele frequencies from paired control populations, or background frequencies observed in European populations from gnomAD [38]. The strongest odds for AAMD conferred collectively by Type 1 *CFI* rare variants was observed in the Kavanagh et. al. (2015) dataset [22] (OR 11.37, 95% CI 10.41–12.74), however this apparent greater enrichment and diversity of genotype may have arisen as a consequence from ascertainment bias, as the study used the same AAMD cohort as Java et. al (2020) [26] who devised the Type 1 *CFI* rare variant genotype list used in this study. Odds of developing AAMD conferred collectively by Type 1 *CFI* rare variants were between 3.1 and 7.8 in the SCOPE, UK Biobank and IAMDGC AAMD datasets. Concordance in ORs derived from the three studies provides good evidence for establishing the level of disease risk, despite variations in AMD population and study designs and which can confound genetic tests of association. As detailed in S1 Table, the studies targeted different forms of AAMD, used different approaches for patient ascertainment and determining patient ethnicity. The stronger ORs observed in the Kavanagh et. al. (2015) [22], SCOPE and FINBB studies may be reflective of the more stringent phenotyping undertaken by retinal specialists (for FINBB this was in a healthcare registry setting), to confirm each patient's AAMD diagnosis and disease stage. This is in comparison to the UK Biobank AAMD study which relied only on an individual's self-reported AMD status to define cases and controls, and reported a weaker OR of 3.1. Another

potential difference was the OR calculations on the SCOPE and FINBB studies that lacked paired controls screened using same method of genotyping, so datasets that matched the underlying ethnicity from gnomAD were used instead. In addition, differences in genotyping strategy employed by each study may have affected prevalence estimates by the ability to detect all protein-altering variation at *CFI*; SCOPE, UK Biobank, FINBB and the Kavanagh et. al. (2015) dataset [22] all used a targeted sequencing approach, but the IAMDGC study [34] relied on a custom modified HumanCoreExome SNP array, with enriched content for protein-altering variants in AAMD risk genes identified in previous targeted next-generation sequencing (NGS) and whole genome sequencing (WGS) studies [42, 48]. Despite the less comprehensive genotyping strategy, the IAMDGC reported comparable collective variant frequencies of Type 1 *CFI* rare variants to the other studies, indicating that there was a good representation of the genotypes on the array.

Collective Type 1 *CFI* rare variant frequencies across different background ethnicities in gnomAD were then investigated [38]. Prevalences varied markedly, with the highest background variant frequency reported in the Ashkenazi Jewish population, followed by the Latino/Admixed American and non-Finnish European populations. This finding aligns with what is known about rare functional variants being restricted to specific populations where historical expansions and contractions can dramatically affect genome-wide patterns of genetic variation in a population [36] which are then further modified by natural selection to amplify or inhibit the frequency of functional alleles [49]. This suggests that Type 1 *CFI* rare variants may have arisen in particular populations which were then kept at low frequencies due to strong negative selection pressures on a tightly regulated CS. Interestingly, a more pronounced population specificity has been described for two rare *CFH* AAMD risk variants, which had highly variable frequencies between different European geographical regions, but the risk estimates for disease attributed to each variant were comparable across the geographical regions studied [50].

Why a higher *CFI* variant prevalence was observedin the background Ashkenazi Jewish population is unclear. Ashkenazi Jews are considered genetically distinct from closely related Europeans and Middle-Eastern populations [51], and have distinct genetic characteristics such as a higher prevalence of common diseases and autosomal recessive diseases [52]. The Ashkenazi Jewish population has previously undergone a historical bottlenecking event, followed by rapid expansion [53]. Deeper inspection showed the *CFI* Type 1 p.Ala240Gly genotype is present in 0.54% background individuals but is nearly absent across any other gnomAD ethnicity, suggesting it may be a founder variant in Ashkenazi Jews (S3 Table). P.Ala240Gly together with p.Gly119Arg account for all the Type 1 *CFI* rare variant Ashkenazi Jewish prevalence rate, and this apparent lack of genetic diversity and expanded p.Ala240Gly frequency fits with the overall picture of a restricted population. Interestingly a different *CFI* rare variant genotype (p.Val412Met), previously linked to low FI levels [25] but not on the prescribed Type 1 list, was reported as the likely disease-causing mutation in three Jewish families affected by early onset of AMD [54]. No studies have investigated yet whether AMD incidence in this population is different to other ethnicities.

A direct comparison was possible between the background collective Type 1 *CFI* rare variant frequency in Finnish population and a dry AMD dataset from the Finnish FINBB network. Type 1 *CFI* rare variants were calculated to confer a greater odds for disease of 8.9 compared to the OR of 3.1 and 7.8 in European AAMD datasets. It is unclear whether a greater disease risk afforded by Type 1 *CFI* rare variants is due to unique factors in the Finnish population, but the reduced background *CFI* variant frequencies in Finnish gnomAD population maybe another consequence of population bottlenecking and genetic drift events known to skew variant frequencies, and increase risk of particular Mendelian diseases [55]. Reasons for why Type

1 *CFI* rare variants confer a greater level of disease risk but are at much lower background frequency and diversity compared to outbred populations, will need to consider the role of environmental or demographic factors in this unique population.

Our study identified a similar collective prevalence of Type 1 *CFI* rare variants in European and Latino/Admixed Americans however the variants were much rarer in African/African-American, Finnish, South Asians and East Asians. These findings illustrate locations for where *CFI* rare variant positive patients may be more likely identified, when considering target population selection for FI therapy. However, before initiating any study these findings should be tested by direct sequencing of AAMD patients from that ethnicity. To determine the feasibility of sequencing studies, we explored whether by using gnomAD frequencies from different background ethnicities, we could predict the sample sizes of AAMD that would be required to detect significant association with Type 1 *CFI* rare variants in those populations that go on to develop AAMD. Power calculations on the Latino/Admixed American and Ashkenazi Jewish background populations identified relatively achievable sample sizes using a targeted sequencing strategy on 1,800 and 670 AAMD individuals from each ethnicity, respectively.

A much greater sample size would technically be required to test for association in South Asians (>15,000 individuals), and sequencing studies in East Asian and African/African-Americans would not be viable given the lack of power even if screening 1,000,000 AAMD individuals. Interestingly, the lack of Type 1 *CFI* rare variants in these populations reflects the trend for lower AMD incidence reported in these ethnicities. Compared to Europeans, incidence of all forms of AMD were lower in Asians (early AMD; 11.2% vs 6.8%, any AMD; 12.3% vs 7.4%), and also lower in Africans (early AMD; 11.2% vs 7.1%, late AMD; 0.5% vs 0.3%, and any AMD; 12.3% vs 7.5%). Europeans also had a higher prevalence of GA compared to Africans, Asians and Hispanics (1.11% vs 0.14%, 0.21% and 0.16%, respectively) [2]. Whether the lower incidence of disease correlates with fewer Type 1 *CFI* rare variants in AAMD in these populations remains an open question. Only by direct sequencing adequately powered AAMD and paired control cohorts from informative non-European populations, can we start to answer these questions and generate the information required to further clinical development of fairer treatments in AAMD.

Our approach makes assumptions that may affect the validity of our findings. Firstly, our chosen European AAMD datasets represent samples selected from the wider affected population and this can introduce sampling errors. This can be reduced by increasing the sample size so it reflects the wider population more effectively, but this comes with additional genotyping costs or may not be technically feasible. Power calculations inform sample sizes required to detect a known association and are especially important when investigating rare variants such as those found in *CFI*. Selection errors, such as when participants in a study 'self-select' to take part, may have also biased the AAMD studies used in our analysis.

The most prevalent Type 1 *CFI* rare variant genotype in our European AMD datasets was p. Gly119Arg, which had an individual rare allele frequency between 0.106% to 1.079%. A previous interaction analysis on p.Gly119Arg frequencies showed no differences in five European contributing datasets in the IAMDGC AAMD GWAS (locations included Eastern USA, Western USA, Western Europe, Britain, and Australia) [50] indicating it is relatively stable across European ethnicities. However, p.Gly119Arg was not detected at comparable levels to Europeans in other gnomAD ethnicities, nor was it detected in 2,119 East Asian neovascular AMD patients and 5,691 paired controls [56]. Focused sequencing across *CFI* in AAMD patients from different ethnicities will determine exactly how confined p.Gly119Arg is to Europeans.

In order to define our target population for *CFI* therapy we relied on the Type 1 *CFI* rare variant list provided by Java et. al. (2020) [26], which provided *in vitro* assessments of each genotype on complement activity. The 18 Type 1 *CFI* rare variant genotypes investigated in

this study were selected on the basis of comprehensive functional analyses showing their correlation with low serum FI levels in carrier blood serum from 29 AAMD patients [26]. Due to the rare nature of these genotypes, and limited availability of *CFI* rare variant positive patient serum samples, only between 1 and 4 samples per genotype were tested *in vitro*. Functional characterisation using high-throughput serum-based functional assays like that described by Java et al. (2020) [26] are required to determine more definitively the clinical impact of rare genotypes identified in affected patients, where *in silico* analysis or prior clinical or functional characterisations are lacking. Despite the limited number of *CFI* rare variant positive samples used previously for functional characterisation, there is strong concordance in findings of FI haploinsufficiency driven by these genotypes in previous studies examining serum from other individuals carrying the nominated Type 1 *CFI* rare variant genotypes [24, 25, 57–60].

Clinically, rare genetic variants are classified according to pathogenicity using ACMG classifications, to support clinical management in the context of rare disease [46]. However, the Type 1 *CFI* rare variants were mostly defined as 'variants of uncertain significance' by ACMG, with only three defined as 'Likely Pathogenic' or 'Pathogenic' (S2 Table), suggesting links between most Type 1 genotypes and pathological phenotype are not well established. P. Gly119Arg was classified as 'Likely Pathogenic' and has been isolated in familial AMD [61] and rare complement disorders [60, 62], but the remainder of genotypes have only been identified in sporadic AMD. How widespread genetic predisposition conferred by Type 1 *CFI* rare variants is within AMD remains unknown, but incomplete penetrance of other genetic variants is widely described across inherited retinal diseases [63]. Challenges in obtaining samples from affected pedigrees, incomplete penetrance, and the late-onset and heterogenous nature of the disease make investigating this challenging.

In addition to Type 1 *CFI* rare variants, Java et al. (2020) also described other *CFI* rare variant genotypes that produce FI proteins with FI levels in the normal range but had varying degrees of reduced enzymatic activity, determined using specialised *in vitro* complement binding assays [26]. Termed 'Type 2' and 'Type 3' *CFI* rare variants, it is plausible that carriers of these genotypes may also benefit from targeted FI gene therapy to restore normal protein function to the retina, rebalancing the CS. However, there is low concordance or a lack of independent verification in the literature for which genotypes meet the 'Type 2' or 'Type 3' categorisation, and less is known about how these genotypes confer elevated risk of AAMD, compared to Type 1 genotypes. For example, the *CFI* rare variant p.Arg406His was categorised as a Type 3 indicating moderate protein dysfunction, but this is contentious in the literature [64, 65]. Additionally, p.Arg406His has been previously reported as protective for AAMD, decreasing risk for disease not increasing risk, like other genotypes (OR: 0.1 [22]). Because of this, prevalences of *CFI* Type 2 and 3 genotypes were not investigated here, but carriers of Type 2 and 3 genotypes remain an important patient sub-group of AMD for the clinical development of FI-targeted therapies.

The lack of available sequencing information from non-European AMD has restricted the field's ability to uncover other rare variants that may be important drivers of disease. Direct sequencing of AMD-related genes in affected patients is required for discovery, and variants significantly associated with disease can then be taken forward to test for replication in independent patients that may or may not share the original genetic background. For clinical development of FI-targeted therapies to reach global populations, there needs to be more sequencing studies conducted on AMD populations from diverse ethnicities and geographies, in order to identify population-specific *CFI* rare variants that can be functionally characterised for their effect on the FI protein, and CS pathway. Any new *CFI* rare variant genotypes identified could be added to the Type 1 (or Type 2 and 3) lists, and individuals with AAMD who carry these variants may ultimately be stratified into clinical trials testing FI targeted

treatments that are seeking to enrol genetically defined sub-populations of AMD. This study describes power calculations used to estimate the likely sample sizes of individuals with AMD from different ethnic backgrounds, that may be required to detect (European) Type 1 *CFI* rare variants if present. However this does not address the sample sizes required to detect functional *CFI* rare variants that might be unique to a different ethnic group, as it is well documented that rare variation is more likely to be restricted to a given population [36, 37]. Furthermore, the sample size analyses performed here were limited to those genetic backgrounds defined by GnomAD, and there are more diverse ethnic groups that remain to be characterised.

Part of the IAMDGC AMD meta-GWAS [34] study design relied on replicating rare protein-altering variants previously identified in a European AAMD cohort in a larger series of AAMD patients largely of European descent [42, 48]. Of note, the IAMDGC study did include samples from Asians (473 cases vs 1,099 controls) and Africans (52 cases vs 361 controls), but due to power limitations, only selected to test association with common (European) AAMD risk SNPs. Whilst this hybrid approach was necessary to study the contribution of rare variation on disease risk across the largest meta-analysis in AAMD to date, it will be critical to the field going forward for large consortia to diversify their patient populations further both at the discovery and replication phases. A statistical process called imputation may provide an alternative approach to informing rare variant prevalences in different ethnicities [66]. However successful imputation relies on using large reference datasets of individuals that share the same ethnic background, and these currently do not exist for non-Europeans and admixed populations, especially at sample sizes required to power accurate imputation of very rare variants like those found in *CFI* [67].

## Why does lack of AMD genetic data from non-European ethnicities matter?

AAMD accounts for 8.7% of all blindness worldwide, especially in people older than 60 years [2]. In the US, the yearly cost for direct health care due to AMD is $255 billion, which is nearly half the direct cost for all vision loss ($513 billion), making AMD the leading cause of visual disability in the developed world and the third globally [68].

Determining the burden and impact to society from AMD is critical for health care planning, but this can only be done by having a precise understanding of disease prevalences. Population-based studies of AMD have indicated differences exist between different racial or ethnic backgrounds, with all forms of AMD being more prevalent in Europeans compared to other ethnicities [2]. This apparent ethnic disparity is also reflected in much lower treatment rates in non-European AMD patients with associated use of anti-VEGF injections, as observed in Medicare claims data [69].

AMD incidences are increasing as a consequence of an expanding aged population; estimations for 2020 indicated there may be 196 million people worldwide suffering with AMD. The number of individuals affected by AMD is projected to rise to 288 million by 2040, with Asia accounting for the largest projected increase in cases, despite having the lowest current estimated prevalence [2]. The US is projected to become more racially and ethnically diverse in the coming years. Growth in admixed populations is already underway, with analysis of the US Census in 2010 predicting that the US will be a 'majority-minority' country by 2060, with African American and Latino populations expanding and admixing significantly [70]. The percentage of the population that is aged 65 years and over is expected to grow from 15% to 24%, equating to an increase of 7% in the native population aged 65 years and over (from 15% to 22%), and a growth of 18% in the foreign-born population (14% to 32%) for the same age group.

Planning for commercialisation of new therapies like those being developed to target FI should consider how changes in population composition may affect a treatment's target population. Understanding genetic prevalences may benefit companies developing therapies by driving inclusion of clinical trial participants that are most likely to benefit from a treatment. Ultimately, this could result in treatments with clinical data and approved labels that can provide information on better real world outcomes across a more diverse patient population than today. This may help close the gap that can arise between results from a registrational trial that demonstrate a treatment is safe and effective, and how that translates to patient outcomes and drug success in a real-world setting [71].

Around 80% of all new drugs are tested in individuals of European descent, and are then administered to individuals of different descent, on the presumption of a generalised therapeutic response [72]. However, similar responses are not guaranteed, for example Asian patients with neovascular AMD reported a much greater rate of a disease presentation with polypoidal choroidal vasculopathy which affected 22 to 55% Asians compared to 5 to 8% European individuals. East Asians with polypoidal choroidal vasculopathy demonstrated a different treatment response to Bevacizumab compared to Europeans [6]. Clinical trial sponsors have traditionally struggled to enrol participants from diverse demographic groups, including racial and ethnic minorities [73], however this may improve following work by the U.S. Food and Drug Administration to better promote the inclusion of underrepresented populations in clinical trials [74]. However, a lot of work still needs to be done to enable evaluation of new and emerging therapies in countries outside of the US and Europe.

Currently, genetic studies in non-European AMD ethnicities are either underpowered or do not use sequencing strategies that enable investigation into how rare variation influences disease [56, 75, 76], or they simply do not exist yet. The current status of AMD genetics reflects the wider landscape in human genetic research, where overrepresentation in genomic databases of Europeans or 'WEIRD' societies (Western, Education, Industrialised, Rich and Democratic) [77], has negative consequences both clinically and ethically [78, 79]. A study on individuals included in the GWAS Catalog, a comprehensive, publicly accessible summary of human genetic association research, revealed that in 2016, 81% of 35 million samples were of European ancestry [80]. Samples from East Asian and South Asian populations accounted for 14%, whereas African, Latin-American, Hispanic, and native or indigenous populations represented less than 4% of all samples analysed. Collectively, these are the most underserved and vulnerable populations in many of the world's richest nations. Under-representation of populations of mixed or non-European ancestry is a problem to precision medicine, as it limits the generalisability of findings from genomics research and reduces the evidence base for translating findings into clinical care for all populations.

The wider genetics community is responding, for example GWASs are now expanding to include greater numbers of individuals from multi-ethnic populations. The U.S. National Institutes of Health mandates the inclusion of diverse participants in the biomedical research it funds. Increased genetic heterogeneity in publicly available research datasets, will require a more comprehensive understanding of the genetic diversity and demographic history to support correct interpretation of results.

## Conclusion

This study determined that the collective odds of AAMD conferred by Type 1 *CFI* rare variants varies between 3.1 to 7.8 in European AAMD datasets, and that disease odds may be even greater in Finnish AAMD (OR 8.9). The lack of available AAMD datasets from other ethnicities prevented exploration of this relationship more globally, so sample sizes were modelled to

illustrate a number that might be required to perform targeted sequencing studies in AAMD cohorts of different ethnicities to detect a significant association between Type 1 *CFI* rare variants and disease risk. This data bias has adverse effects on clinical development and future commercialisation strategies for targeted treatments in AAMD, which has a vast unmet and expanding need. The conclusions of this study further reinforce the need to generate increasingly diverse genetic AAMD datasets to identify the contribution of genomic variants more accurately, to support development of successful treatments for all people affected by this devastating disease.

## Supporting information

**S1 Table. Description of European AAMD and control datasets used to evaluate prevalence of Type 1 *CFI* rare variants.**
(DOCX)

**S2 Table. Type 1 *CFI* rare variant frequencies in different European AAMD and control datasets.**
(DOCX)

**S3 Table. Type 1 *CFI* rare variant frequencies in different gnomAD populations.**
(DOCX)

**S4 Table. Diversity of Type 1 *CFI* rare variant genotypes observed in background gnomAD populations.**
(DOCX)

## Acknowledgments

The authors thank all the participants who took part in the SCOPE, UK Biobank, FINBB, IAMDGC and gnomAD studies, and support staff who made this possible. Finnish biobanks [Auria Biobank, Finnish Clinical Biobank Tampere, Helsinki Biobank, Biobank of Eastern Finland, Biobank of Central Finland, Biobank Borealis] are acknowledged for having provided samples and data to this study. The Fingenious® digital gateway (www.fingenious.fi) by the Finnish Biobank Cooperative—FINBB provides access to Finnish biobank services. The authors thank Dr Hyung-Woo Kim from Gyroscope Therapeutics for statistical support, and Tiffany Howard and Martin Quinn from Gyroscope Therapeutics for assistance in providing critical comments on the manuscript. The authors declare the following competing interests: Drs Jones, Curtiss, Harris and Waheed are employees of Gyroscope Therapeutics Limited.

### SCOPE study group membership

Dr Rashi Arora
    Salisbury NHS Foundation Trust
    Wilshire, UK
    Dr Courtney Crawford
    Strategic Clinical Research Group LLC
    Willow Park, USA
    Dr Catherine Creuzot-Garcher
    CHU Dijon—Hopital Mitterrand
    Cedex, France
    Dr Karl Csaky
    Retina Foundation of the Southwest

Dallas, USA
Dr Francois Devin
Centre Paradis Monticelli
Marseille, France
Dr David Eichenbaum
Retina Vitreous Associates of Florida
St. Petersburg, USA
Dr Philip Ferrone
Long Island Vitreoretinal Consultants
Great Neck, USA
Dr Marta Figueroa
VISSUM Mirasierra
Madrid, Spain
Dr Christina Flaxel
Casey Eye Institute–OHSU
Portland, USA
Dr Ghassan Ghorayeb
West Virginia University
Morgantown, USA
Dr David Gilmour
NHS Greater Glasgow and Clyde
Glasgow, UK
Dr Salvatore Grisanti
University Hospital Schleswig-Holstein Campus
Luebeck, Germany
Prof. Robyn Guymer
The University of Melbourne—The Centre for Eye Research Australia (CERA)
East Melbourne, Australia
Dr Edward Hall
Retina Association of Western New York
Rochester, USA
Dr Jeff Heier
Ophthalamic Consultants of Boston (OCB)
Boston, USA
Dr Allen Ho
Mid-Atlantic Retina
Huntingdon Valley, USA
Prof. Carol Hoyng
Radboud Universitair Medisch Centrum
Nijmegen, Netherlands
Dr Peter Charbel Issa
John Radcliffe Hospital
Oxford, UK
Dr Tsveta Ivanova
Manchester Royal Eye Hospital
Manchester, UK
Dr Bartlomiej Kaluzny
Oftalmika—Prywatna Klinika Okulistyczna
Bydgoszcz, Poland

Dr Ashad Khanani
Sierra Eye Associates
Reno, USA
Dr Nicolas Leveziel
Centre Hospitalier Universitaire de Poitiers—Poitiers University Hospital
Poitiers Cedex, France
Dr Raj Maturi
Midwest Eye Institute Northside
Indianapolis, USA
Dr Martin McKibbin
Leeds Teaching Hospitals NHS Trust
Leeds, UK
Dr Jared Nielsen
Wolfe Eye Clinic
West Des Moines, USA
Dr Todd Schneiderman
Retina Center Northwest
Silverdale, USA
Dr Martin Spitzer
Universitaetsklinikum Hamburg-Eppendorf
Hamburg, Germany
Prof. David Steele
Sunderland Eye Infirmary
Sunderland, UK
Dr Eric Suan
The Retina Care Center
Baltimore, USA
Dr Tongalp H. Tezel (SCOPE study lead author)
Columbia University Medical Center
New York, USA
Contact email address: Tht2115@cumc.columbia.edu
Dr Vinod Voleti
New Jersey Retina Research Foundation
Great Neck, USA
Dr Robert Wirthlin
Spokane Eye Clinical Research
Spokane, USA

## Author Contributions

**Conceptualization:** Amy V. Jones, Nadia K. Waheed.

**Data curation:** Amy V. Jones, Tom Southerington, Marco Hautalahti, Pauli Wihuri, Johanna Mäkelä, Roosa E. Kallionpää, Enni Makkonen, Theresa Knopp, Arto Mannermaa, Erna Mäkinen, Anne-Mari Moilanen.

**Formal analysis:** Amy V. Jones.

**Funding acquisition:** Nadia K. Waheed.

**Investigation:** Amy V. Jones.

**Methodology:** Amy V. Jones.

**Project administration:** Amy V. Jones.

**Writing – original draft:** Amy V. Jones.

**Writing – review & editing:** Amy V. Jones, Darin Curtiss, Claire Harris, Tongalp H. Tezel.

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
