## [Decision Letter · Decision Letter 0]

27 Apr 2022

PONE-D-22-04791How does ethnicity affect prevalence of AMD-associated CFI rare genetic variants, and why does it matter?PLOS ONE

Dear Dr. Waheed,

Thank you for submitting your manuscript to PLOS ONE. After careful consideration, we feel that it has merit but does not fully meet PLOS ONE’s publication criteria as it currently stands. Therefore, we invite you to submit a revised version of the manuscript that addresses the points raised during the review process.

We look forward to receiving your revised manuscript.

Kind regards,

Anand Swaroop

Academic Editor

PLOS ONE

Journal Requirements:

2. For studies reporting research involving human participants, PLOS ONE requires authors to confirm that this specific study was reviewed and approved by an institutional review board (ethics committee) before the study began. Please provide the specific name of the ethics committee/IRB that approved your study, or explain why you did not seek approval in this case.

3. Thank you for stating the following in the Competing Interests: 

(The authors have declared that no competing interests exist:

Tom Southerington

Enni Makkonen

Roosa E. Kallionpää

Pauli Wihuri

Theresa Knopp 

Marco Hautalahti

Johanna Mäkelä

Arto Mannermaa

Erna Mäkinen

Anne-Mari Moilanen

Authors with competing interests

I have read the journal's policy and the authors of this manuscript have the following competing interests: 

Amy Jones: employee and share holder of Gyroscope Therapeutics Ltd

Darin Curtiss: employee and share holder of Gyroscope Therapeutics Ltd

Claire Harris: employee and share holder of Gyroscope Therapeutics Ltd, Research grant from RA Phaemaceutics (payment to institution), Royalty income from commercialized factor I ELISA; Hycult Biotech, consultancy income from Q32 Bio Inc, Chinook Therapeutics, and Biocryst Pharmaceuticals (all payment to institution), 

Nadia Waheed: employee and share holder of Gyroscope Therapeutics Ltd, grants from Carl Zeiss Meditec, Topcon, Regeneron, Heidelberg, Nidek, Optovue, consultancy income from Apellis, Nidek, Boehringer Ingelheim, stock in Ocudyne.)

We note that one or more of the authors have an affiliation to the commercial funders of this research study : Gyroscope Therapeutics Ltd, Research grant from RA Phaemaceutics (payment to institution), Hycult Biotech, Q32 Bio Inc, Chinook Therapeutics, Biocryst Pharmaceuticals

Within your Competing Interests Statement, please confirm that this commercial affiliation does not alter your adherence to all PLOS ONE policies on sharing data and materials by including the following statement: ""This does not alter our adherence to  PLOS ONE policies on sharing data and materials.” (as detailed online in our guide for authors http://journals.plos.org/plosone/s/competing-interests). If this adherence statement is not accurate and  there are restrictions on sharing of data and/or materials, please state these. Please note that we cannot proceed with consideration of your article until this information has been declared.

4. One of the noted authors is a group or consortium (Scope Study Group). In addition to naming the author group, please list the individual authors and affiliations within this group in the acknowledgments section of your manuscript. Please also indicate clearly a lead author for this group along with a contact email address.

Additional Editor Comments:

The two reviewers have raised important points of concerns and clarification. The key issue is that despite the title of effect of "ethnicity" no non-European cohort was included. In addition, the impact of these studies requires better clarification.

Reviewers' comments:

Reviewer's Responses to Questions

**Comments to the Author**

1. Is the manuscript technically sound, and do the data support the conclusions?

Reviewer #1: Yes

Reviewer #2: Partly

2. Has the statistical analysis been performed appropriately and rigorously? 

Reviewer #1: Yes

Reviewer #2: Yes

3. Have the authors made all data underlying the findings in their manuscript fully available?

Reviewer #1: Yes

Reviewer #2: Yes

4. Is the manuscript presented in an intelligible fashion and written in standard English?

Reviewer #1: Yes

Reviewer #2: Yes

5. Review Comments to the Author

Reviewer #1: The paper focuses on a list of 18 rare variants in CFI associated with AMD, and investigating their frequencies in several populations, using genetic data sets from different organizations and countries.

The lack of genetic diversity, and the implication of this lack of information is a topic of discussion and investigation by many in the field of human genetics. At this point, it is a well-established fact that lack of proper genetic controls and background distribution hinders our understanding of human disease development, as well as development of treatment.

I welcome and agree with the sentiments of the authors on the immense need of a more genetically diverse data bases to be able accurately describe the genetic variability as well as disease linked variants in a given population (in this case rare variants contributing to AMD). With that being said, I feel that the authors focus most of the paper on the well described and curated European populations. The title of the paper is "How does ethnicity affect prevalence of AMD-associated CFI rare genetic variants, and why does it matter?" but throughout the paper the authors talk about a select list of variants that was selected based on European population and the other ethnicities are summarized in to we don't have enough data to show an association. I think that the calculations about the needed numbers of affected are a good starting point, but this will not be sufficient until the list of variants to be investigated is also matched to the population in question. For example, they mention the reported p.V412M variant in CFI identified in 3 Tunisian Jewish families, this is a different population from the Ashkenazi Jewish mentioned in the paper, so it should be investigated separately (or compared to other Northern African diaspora populations), so it is strange to me that it was noted together with the Ashkenazi analysis. The same paper also reported a CFI variant found in an Ashkenazi family (p.K441R) but this was not included in the analysis. The prevalent variants in each population may be very different so accounting for variants found in population A to investigate association in population B will always be an incomplete analysis.

I think that the part dedicated to European analysis should be significantly shorter, the contribution of CFI to AMD is not new to the field, and there is no major new findings there from my perspective. If more ethnicities can be added to the mix, I think this will be more informative, and can be used to guide organizations like the International Age-related Macular Degeneration Genomics Consortium (IAMDGC) and EYE-RISK in research design and focus.

There is also a place to mention that some of the variants described in the paper are not rare in all population, which should also be taken into account when we discuss the causality or pathogenicity of these variants. This is not necessarily within the scope of this paper, but I do think it should be mentioned.

Reviewer #2: This manuscript focuses on understanding the role of rare variants in CFI on Age-related Macular Degeneration across different ethnicities (AMD). Both common and rare variants in CFI, which is a complete gene, have been associated with the risk of AMD. The authors focus on the rare variants that have been shown in earlier studies to affect the levels of CFI in serum and analyze them across multiple datasets. They find enrichment of CFI variants across all European cohorts and estimate the sample size needed to access the rare variants across different ethnicities. This reviewer has the following comets that I would like the authors to address:

1.The authors provide very little background on the 18 Types 1 variants analyzed in this study. It will be good to provide brief information on how many samples were analyzed, and what kind of sequencing was performed (exome or targeted sequencing) and what proportion of patients had rare variants in CFI.

2.Based on Java et al., 2020 Type 1 variants were defined as those that demonstrated low FI antigenic levels and low iC3b generation, and Type 2 and 3 variants that exhibited normal levels but reduced function. Both of these categories can have an impact on the AMD disease outcome. Then why did the authors choose to follow only type-1 variants?

3.While the UK biobank, SCOPE, and genomeAD database have sequencing data, IAMGDC data was based on the genotypes. They did include the rare variants, but it was not as comprehensive as sequencing-based discovery and is limited by the variants present in the array. This should be clearly stated.

4.The odds ratio for type-1 variants is high and closer to the original analysis Kavanagh et al., 2015. Could the authors offer an explanation for this finding? Could it be attributed to better phenotyping of the AMD and controls in IAMDGC and Kavanagh et al? I would like the authors to include this in the discussion.

5.The focus of the paper centers around finding the difference in the frequency of the rare variant based on ethnicity. This is well-documented that rare variants tend to be specific to the population. All of the data presented here are from the publicly available data which shows the frequency differences across populations, which is not surprising. Authors then predict the estimate of the sample size needed to study type I variants across different ethnicity. However, the motivation for this is not clear. Different ethnicities are likely to have their unique, rare variants. It is also likely that in some populations the disease may not be driven by the rare variants in CFI. Could the authors explain the rationale behind this?

6. PLOS authors have the option to publish the peer review history of their article (what does this mean?). If published, this will include your full peer review and any attached files.

Reviewer #1: No

Reviewer #2: No

---

## [Author Response · Author response to Decision Letter 0]

13 Jun 2022

Rebuttal letter

Manuscript number: PONE-D-22-04791

Date: 13th May 2022

We thank the editor and the two reviewers for their comments on our manuscript. Below is

our response to each point raised by the academic editor and reviewers. We hope that we

satisfyingly addressed them and that the manuscript will be now suited for publication.

Sincerely, 

On behalf of all authors,

Dr Amy V. Jones

…………………………………………………………………………………………………………………………………………………

Editorial comments:

Journal Requirements:

Based on the PLOS ONE’s style requirements, we have fixed the section header formatting, bolding of supplemental table citation format and removed the abbreviations in the co-authors affiliations.

2. For studies reporting research involving human participants, PLOS ONE requires authors to confirm that this specific study was reviewed and approved by an institutional review board (ethics committee) before the study began. Please provide the specific name of the ethics committee/IRB that approved your study, or explain why you did not seek approval in this case.

Author response:

The Methods Ethics Statement has now been updated to provide the ethics committees that reviewed and approved the SCOPE and Finnish studies, which represent the novel datasets in this manuscript. Additionally, we also describe summary results from previously published studies, and now include reference to those publications that detail the ethics review committees/IRBs that reviewed and approved previously published datasets (lines 269 to 290). 

3. Thank you for stating the following in the Competing Interests: 

(The authors have declared that no competing interests exist:

Tom Southerington

Enni Makkonen

Roosa E. Kallionpää

Pauli Wihuri

Theresa Knopp 

Marco Hautalahti

Johanna Mäkelä

Arto Mannermaa

Erna Mäkinen

Anne-Mari Moilanen

Authors with competing interests

I have read the journal's policy and the authors of this manuscript have the following competing interests: 

Amy Jones: employee and share holder of Gyroscope Therapeutics Ltd

Darin Curtiss: employee and share holder of Gyroscope Therapeutics Ltd

Claire Harris: employee and share holder of Gyroscope Therapeutics Ltd, Research grant from RA Phaemaceutics (payment to institution), Royalty income from commercialized factor I ELISA; Hycult Biotech, consultancy income from Q32 Bio Inc, Chinook Therapeutics, and Biocryst Pharmaceuticals (all payment to institution), 

Nadia Waheed: employee and share holder of Gyroscope Therapeutics Ltd, grants from Carl Zeiss Meditec, Topcon, Regeneron, Heidelberg, Nidek, Optovue, consultancy income from Apellis, Nidek, Boehringer Ingelheim, stock in Ocudyne.)

We note that one or more of the authors have an affiliation to the commercial funders of this research study : Gyroscope Therapeutics Ltd, Research grant from RA Phaemaceutics (payment to institution), Hycult Biotech, Q32 Bio Inc, Chinook Therapeutics, Biocryst Pharmaceuticals

Author response:

We have revised the Funding Statement to state ‘Gyroscope Therapeutics Limited provided funding for the design and conduct of the SCOPE study, and also provided funding, approved the design, conduct, preparation, analysis and interpretation of the data presented in this manuscript, and decision to submit the manuscript for publication. This study was performed as part of the authors’ regular employment duties and the funder provided support in the form of salaries for authors (AJ, DC, CH, NW). (Lines 801-806).

Within your Competing Interests Statement, please confirm that this commercial affiliation does not alter your adherence to all PLOS ONE policies on sharing data and materials by including the following statement: ""This does not alter our adherence to PLOS ONE policies on sharing data and materials.” (as detailed online in our guide for authors http://journals.plos.org/plosone/s/competing-interests). If this adherence statement is not accurate and there are restrictions on sharing of data and/or materials, please state these. Please note that we cannot proceed with consideration of your article until this information has been declared.

Author response:

We have duly revised the cover letter to include a Funding Statement and Competing Interests Statement, and included text from the response to 2a above, and the sentence ‘The commercial interest described above does not alter our adherence to all PLOS ONE policies on data sharing and materials.’ (Lines 801-806).

Author response:

We included the Data Availability statement ‘Yes - all data are fully available without restriction,’ but this did not state we would provide this via a repository. All novel data presented are all contained within the manuscript and/or Supporting Information files already provided, and to this end we provided the statement ‘All relevant data are within the manuscript and its Supporting Information files.’ 

Therefore we understand that there are no changes to the existing Data Availability statement. Please advise if there needs to be further clarification. 

4. One of the noted authors is a group or consortium (Scope Study Group). In addition to naming the author group, please list the individual authors and affiliations within this group in the acknowledgments section of your manuscript. Please also indicate clearly a lead author for this group along with a contact email address.

Author response:

A lead author for the SCOPE study is now included in the authorship list, and a contact email address is provided in the Supportive Information section (line 797). 

Additional Editor Comments:

The two reviewers have raised important points of concerns and clarification. The key issue is that despite the title of effect of "ethnicity" no non-European cohort was included. In addition, the impact of these studies requires better clarification.

Reviewers' comments:

Reviewer's Responses to Questions

Comments to the Author

1. Is the manuscript technically sound, and do the data support the conclusions?

Reviewer #1: Yes

Reviewer #2: Partly

2. Has the statistical analysis been performed appropriately and rigorously?

Reviewer #1: Yes

Reviewer #2: Yes

3. Have the authors made all data underlying the findings in their manuscript fully available?

Reviewer #1: Yes

Reviewer #2: Yes

4. Is the manuscript presented in an intelligible fashion and written in standard English?

Reviewer #1: Yes

Reviewer #2: Yes

5. Review Comments to the Author

Reviewer #1: The paper focuses on a list of 18 rare variants in CFI associated with AMD, and investigating their frequencies in several populations, using genetic data sets from different organizations and countries.

The lack of genetic diversity, and the implication of this lack of information is a topic of discussion and investigation by many in the field of human genetics. At this point, it is a well-established fact that lack of proper genetic controls and background distribution hinders our understanding of human disease development, as well as development of treatment.

I welcome and agree with the sentiments of the authors on the immense need of a more genetically diverse data bases to be able accurately describe the genetic variability as well as disease linked variants in a given population (in this case rare variants contributing to AMD). With that being said, I feel that the authors focus most of the paper on the well described and curated European populations. The title of the paper is "How does ethnicity affect prevalence of AMD-associated CFI rare genetic variants, and why does it matter?" but throughout the paper the authors talk about a select list of variants that was selected based on European population and the other ethnicities are summarized in to we don't have enough data to show an association. I think that the calculations about the needed numbers of affected are a good starting point, but this will not be sufficient until the list of variants to be investigated is also matched to the population in question. For example, they mention the reported p.V412M variant in CFI identified in 3 Tunisian Jewish families, this is a different population from the Ashkenazi Jewish mentioned in the paper, so it should be investigated separately (or compared to other Northern African diaspora populations), so it is strange to me that it was noted together with the Ashkenazi analysis. The same paper also reported a CFI variant found in an Ashkenazi family (p.K441R) but this was not included in the analysis. The prevalent variants in each population may be very different so accounting for variants found in population A to investigate association in population B will always be an incomplete analysis.

I think that the part dedicated to European analysis should be significantly shorter, the contribution of CFI to AMD is not new to the field, and there is no major new findings there from my perspective. If more ethnicities can be added to the mix, I think this will be more informative, and can be used to guide organizations like the International Age-related Macular Degeneration Genomics Consortium (IAMDGC) and EYE-RISK in research design and focus.

There is also a place to mention that some of the variants described in the paper are not rare in all population, which should also be taken into account when we discuss the causality or pathogenicity of these variants. This is not necessarily within the scope of this paper, but I do think it should be mentioned.

Author response:

We thank reviewer 1 for their comments that this manuscript highlights an important gap in the genomics data landscape currently, that the lack of non-European genetic datasets hampers our global understanding around the contribution of rare variants to common diseases, here we discuss AMD as a good example. Due to the absence of genetic data from individuals with AMD from non-European background, we were unable to answer the question about how rare variation in the CFI gene affects AMD in global populations. The aim of the paper was to provide prevalence estimates for CFI rare variants associated with AMD in non-overlapping European AMD datasets, highlight the disparity that this assessment cannot be done in non-European datasets, and discuss why this is important for healthcare equity and developing new treatments for this devastating disorder. 

To better explain the intention of the manuscript, we updated the manuscript title to better signal the CFI rare variants were definable only in European populations; ‘An assessment of prevalence of Type 1 CFI rare variants in AMD, and why lack of genetic data from non-European ethnicities hinders development of new treatments and healthcare access’. 

The main novelty is the description of the frequency prevalences of rare CFI variant genotypes associated with low serum FI levels (referred to as ‘Type 1’), in two newly described AMD cohorts derived from the SCOPE natural history study, and a Finnish Biobank study (FINBB). Combined with prevalences derived from previously published AMD datasets, some of which required secondary analyses to determine (e.g. UK Biobank), this manuscript presents a comprehensive overview of the incidence of Type 1 rare CFI variants in European AMD. Understanding the size of this AMD sub-population has direct implications for the development of FI targeted gene therapies, and identifying target patient populations globally. 

In the results section, we present collective prevalences for five different European AMD datasets, two of which are novel (SCOPE and FINBB). There are currently no published comparable genetic datasets in any non-European ethnicity as far as we are aware from extensive searches on Pubmed. Therefore to answer the reviewers question, deriving comparable prevalence estimates in Ashkenazi Jewish, or Tunisian Jewish sporadic AMD populations, is not currently possible. We cite in the discussion that there have been a small number of publications describing sequencing studies in familial AMD from both these ethnic groups that have isolated rare CFI variants in affected, related individuals. These studies were restricted to investigating only a small number of families, and do not provide a full picture of the breadth of CFI rare variation that may exist in individuals with sporadic AMD. Later in the manuscript, we predict the sample sizes that might be required for screening to determine prevalence of CFI rare variants in AMD arising from the closest ethnicities to these groups, using background (control) datasets from gnomAD. As the reviewer explains, this could serve as a guide for organisations like IAMDGC and EYE-RISK for future research design and focus. 

Lastly, reviewer 1 states that there is ‘a place to mention that some of the variants described in the paper are not rare in all population’. The 18 CFI rare variant genotypes (as listed in S2 and S3 Tables) all individually have minor allele frequencies equal or less than 1% in background (European) populations, including non-Finnish Europeans from gnomAD. We believe this terminology using ‘rare’ in this context to be broadly correct as it aligns with the current definitions that rare variants are alternative forms of a gene that are present with a minor allele frequency (MAF) of less than 1%. However we acknowledge that the manuscript can be improved by better qualifying how we are using the term ‘rare’, therefore we added the following statement into the Methods section:

‘The term ‘rare’ refers throughout to variants that are equal or less than 1% minor allele frequency (MAF) in European background (control) populations (Cirulli and Goldstein Nat Rev Genet. 2010 Jun;11(6):415-25. doi: 10.1038/nrg2779).’ Lines 180-182. 

Reviewer #2: This manuscript focuses on understanding the role of rare variants in CFI on Age-related Macular Degeneration across different ethnicities (AMD). Both common and rare variants in CFI, which is a complete gene, have been associated with the risk of AMD. The authors focus on the rare variants that have been shown in earlier studies to affect the levels of CFI in serum and analyze them across multiple datasets. They find enrichment of CFI variants across all European cohorts and estimate the sample size needed to access the rare variants across different ethnicities. This reviewer has the following comets that I would like the authors to address:

1.The authors provide very little background on the 18 Types 1 variants analyzed in this study. It will be good to provide brief information on how many samples were analyzed, and what kind of sequencing was performed (exome or targeted sequencing) and what proportion of patients had rare variants in CFI.

Author response:

The patient numbers for those carrying rare CFI variant genotypes identified as Type 1 in the Java et al. (2020) paper are not representative of the overall population frequency because this study only cites results from in vitro testing of patient samples, which were selected on the basis of serum blood sample availability. Instead, we cited an earlier study (Kavanagh et al. (2015)) that describes CFI rare variant prevalences derived from the same underlying AMD patient dataset, and additionally from a matched control cohort (Table 1). The definition of Type 1 for the CFI rare variant genotypes analysed in vitro by Java et al. (2015) were on a limited number of samples. However the finding of low FI levels (‘haploinsufficiency’) in carrier blood serum accords with that from many previous observations reported in the literature, boosting confidence that these genotypes are functional. 

We have revised the text in the Methods and Discussion to explain these points more clearly, as follows:

Methods:

For CFI variant frequency analysis, we selected the 18 ‘Type 1’ CFI rare variant genotypes defined by Java et. al. (2020) (25), in a European AAMD cohort, on the basis 29 carrier patient serum blood samples underwent comprehensive serum-based in vitro functional assays measuring FI level and complement activity to determine ‘Type 1’ status. 

Because Java et. al. (2020) described the functional nature of different CFI rare variant from only in a minority of cases with serum available for functional evaluation of each genotype, more accurate variant frequency data was sourced from an earlier study examining the same underlying European AAMD (n=2,266) and control sample (n=1,400) datasets, as reported by Kavanagh et. al. (2015) (21), which employed a targeted next-generation sequencing strategy to capture the CFI coding and 5’ untranslated (UTR) and 3’ UTR regions (Table 1) (21). Lines 161-170.

Discussion:

The 18 Type 1 CFI rare variant genotypes investigated in this study were selected on the basis of comprehensive functional analyses showing their correlation with low serum FI levels in carrier blood serum (25). Due to the rare nature of these genotypes, and limited availability of CFI rare variant positive patient serum samples, only between 1 and 4 samples per genotype were tested in vitro. Functional characterisation using high-throughput serum-based functional assays like that described by Java et al. (2020) (25) are required to determine more definitively the clinical impact of rare genotypes identified in affected patients, where in silico analysis or prior clinical or functional characterisations are lacking. Despite the limited number of CFI rare variant positive samples used previously for functional characterisation, there is strong concordance in findings of FI haploinsufficiency driven by these genotypes in previous studies examining serum from other individuals carrying the nominated Type 1 CFI rare variant genotypes (23,24,55–58). Lines 514-524.

2.Based on Java et al., 2020 Type 1 variants were defined as those that demonstrated low FI antigenic levels and low iC3b generation, and Type 2 and 3 variants that exhibited normal levels but reduced function. Both of these categories can have an impact on the AMD disease outcome. Then why did the authors choose to follow only type-1 variants?

Author response:

Those CFI rare variant genotypes categorised as Type 1 are considered to have the most direct functional impact by reducing FI levels systemically, and carriers of these genotypes who are affected by AMD are hypothesised to be the sub-population of patients who would maximally benefit from FI-targeted gene therapy. We acknowledge that other CFI rare variant genotypes, categorised as Type 2 and 3, also impact negatively by reducing protein function but not levels. However, establishing a dysfunctional consequence from Type 2 and 3 genotypes on resulting FI protein requires specialised complement binding assays. This has made the functional consequence of Type 2 and 3 genotypes more challenging to characterise than Type 1 genotypes, where measuring FI serum levels is technically more straightforward. Because of this, there is a greater body of evidence in the literature supporting correlations between Type 1 genotypes and (low) FI level. The level of risk of AAMD conferred by Type 1 CFI rare variant genotypes has been more deeply characterised compared to Type 2 and 3 CFI rare variants, and (limited) data suggests Type 1 genotypes may confer a greater magnitude of disease risk, further confirming Type 1 genotypes as being more characterised in terms of their contribution to disease. 

To address this point raised by Reviewer 2 in the manuscript, we added the following paragraph to the discussion:

In addition to Type 1 CFI rare variants, Java et al. (2020) also described other CFI rare variant genotypes that produce FI proteins with FI levels in the normal range but had varying degrees of reduced enzymatic activity, determined using specialised in vitro complement binding assays (Ref). Termed ‘Type 2’ and ‘Type 3’ CFI rare variants, it is plausible that carriers of these genotypes may also benefit from targeted FI gene therapy to restore normal protein function to the retina, rebalancing the CS. However, there is low concordance or a lack of independent verification in the literature for which genotypes meet the ‘Type 2’ or ‘Type 3’ categorisation, and less is known about how these genotypes confer elevated risk of AAMD, compared to Type 1 genotypes. For example, the CFI rare variant p.Arg406His was categorised as a Type 3 indicating moderate protein dysfunction, but this is contentious in the literature (64,65). Additionally, p.Arg406His has been previously reported as protective for AAMD, decreasing risk for disease not increasing risk, like other genotypes (OR: 0.1 (22). Because of this, prevalences of CFI Type 2 and 3 genotypes were not investigated here, but carriers of Type 2 and 3 genotypes remain an important patient sub-group of AMD for the clinical development of FI-targeted therapies. Lines 536-549. 

3.While the UK biobank, SCOPE, and genomeAD database have sequencing data, IAMGDC data was based on the genotypes. They did include the rare variants, but it was not as comprehensive as sequencing-based discovery and is limited by the variants present in the array. This should be clearly stated.

Author response:

We have updated the discussion to better call out the differences in the sequencing strategies between the studies as suggested by Reviewer 2, by adding a new paragraph:

In addition, differences in genotyping strategy employed by each study may have affected prevalence estimates by the ability to detect all protein-altering variation at CFI; SCOPE, UK Biobank, FINBB and the Kavanagh et. al. (2015) dataset (21) all used a targeted sequencing approach, but the IAMDGC study (33) relied on a custom modified HumanCoreExome SNP array, with enriched content for protein-altering variants in AAMD risk genes identified in previous targeted next-generation sequencing (NGS) and whole genome sequencing (WGS) studies (41,47). Despite the less comprehensive genotyping strategy, the IAMDGC reported comparable collective variant frequencies of Type 1 CFI rare variants to the other studies, indicating that there was a good representation of the genotypes on the array. Lines 424-432. 

We have also updated S1 Table with an additional column to describe the methods for variant identification utilised in each AAMD study. 

4.The odds ratio for type-1 variants is high and closer to the original analysis Kavanagh et al., 2015. Could the authors offer an explanation for this finding? Could it be attributed to better phenotyping of the AMD and controls in IAMDGC and Kavanagh et al? I would like the authors to include this in the discussion.

Author response:

Since this manuscript was created we have performed a new datacut on the SCOPE study, and the sample size has increased to 3,243 patients with the odds ratio (OR) for AAMD increasing to 7.8. We believe this greater sample size improves the manuscript; therefore we have taken the opportunity to revise the SCOPE data presented in Table 1 and S2 Table with the updated frequencies, and we have updated the values cited throughout the manuscript. 

This update of the SCOPE study OR to 7.8 brings it closer to that reported by the IAMDGC and Kavanagh et. al. (2015) studies, who reported ORs of 4.2 and 11.4, respectively. We agree that the stronger ORs may be reflective of the more stringent phenotyping of AMD performed by retinal specialists, in comparison to the UK Biobank AMD study that reported a weaker OR of 3.1, which only relied on self-reported AMD status. We have updated the manuscript discussion to describe this:

The stronger ORs observed in the Kavanagh et. al. (2015) (21), SCOPE and FINBB studies may be reflective of the more stringent phenotyping undertaken by retinal specialists (for FINBB this was in a healthcare registry setting), to confirm each patient’s AAMD diagnosis and disease stage. This is in comparison to the UK Biobank AAMD study which relied only on an individual’s self-reported AMD status to define cases and controls, and reported a weaker OR of 3.1. Lines 417-422. 

5.The focus of the paper centers around finding the difference in the frequency of the rare variant based on ethnicity. This is well-documented that rare variants tend to be specific to the population. All of the data presented here are from the publicly available data which shows the frequency differences across populations, which is not surprising. Authors then predict the estimate of the sample size needed to study type I variants across different ethnicity. However, the motivation for this is not clear. Different ethnicities are likely to have their unique, rare variants. It is also likely that in some populations the disease may not be driven by the rare variants in CFI. Could the authors explain the rationale behind this?

Author response:

One of the key motivations for this study was exploring the prevalence of Type 1 CFI rare variants linked to FI protein haploinsufficiency, in non-European AMD datasets. Characterising this subgroup of AMD patients is important for clinical development and commercialisation of targeted FI therapies, as AMD patients carrying Type 1 CFI rare variants are hypothesised to be most likely to benefit. These analyses try to define the size and distribution of the (Type 1) target populations that might exist, outside of European AMD. But as there is a lack of genetic data in AMD from non-Europeans, we are unable currently to answer this question. To try to drive the field to address this, we present power calculations for possible sample sizes of AMD patients from different non-European backgrounds that would need to be screened prospectively to determine wither the Type 1 CFI rare variants previously characterised, are present or likely absent. 

Of course, this does not preclude there being other CFI rare variant genotypes that are unique to a given non-European AMD population, which may be associated with low serum FI levels and cause complement dysregulation. However these genotypes firstly need to be identified by new studies, then functionally characterised to prove a robust association with low FI levels, before they can be used to direct patient stratification to clinical trials testing FI-targeted therapies that are seeking to enrol genetically defined sub-populations. To better describe this, we added to the introduction and discussion:

Introduction:

This may stimulate the field to generate more sequencing studies in individuals with AAMD from diverse ethnic backgrounds, further supporting the clinical development of novel therapies which could reach a wider range of AAMD patients. Lines 151-153

Discussion:

For clinical development of FI-targeted therapies to reach global populations, there needs to be more sequencing studies conducted on AMD populations from diverse ethnicities and geographies, in order to identify population-specific CFI rare variants that can be functionally characterised for their effect on the FI protein, and CS pathway. Any new CFI rare variant genotypes identified could be added to the Type 1 (or Type 2 and 3) lists, and individuals with AAMD who carry these variants may ultimately be stratified into clinical trials testing FI targeted treatments that are seeking to enrol genetically defined sub-populations of AMD. This study describes power calculations used to estimate the likely sample sizes of individuals with AMD from different ethnic backgrounds, that may be required to detect (European) Type 1 CFI rare variants if present. However this does not address the sample sizes required to detect functional CFI rare variants that might be unique to a different ethnic group, as it is well documented that rare variation is more likely to be restricted to a given population (35,36). Furthermore, the sample size analyses performed here were limited to those genetic backgrounds defined by GnomAD, and there are more diverse ethnic groups that remain to be characterised. Lines 554-567. 

AAMD is a complex disorder, with multiple risk factors that include genetics but also non-genetic factors, such as age, environment, diet and sunlight exposure. We have updated the Introduction to better describe the interplay between genetics and environment:

AMD is a complex disease with a multifactorial aetiology influenced by age and environmental factors such as diet and sunlight exposure, and a positive family history (3,5). Lines 93-95.

---

## [Decision Letter · Decision Letter 1]

6 Jul 2022

PONE-D-22-04791R1An assessment of prevalence of Type 1 CFI rare variants in AMD, and why lack of genetic data from non-European ethnicities hinders development of new treatments and healthcare accessPLOS ONE

Dear Dr. Waheed,

Thank you for submitting your manuscript to PLOS ONE. After careful consideration, we feel that it has merit but does not fully meet PLOS ONE’s publication criteria as it currently stands. Therefore, we invite you to submit a revised version of the manuscript that addresses the points raised during the review process.

We look forward to receiving your revised manuscript.

Kind regards,

Anand Swaroop

Academic Editor

PLOS ONE

Journal Requirements:

Additional Editor Comments:

In response to the Reviewer 2's continued concern about the lack of non-European populations in the manuscript, the authors should modify the text appropriately to avoid any over-reach or over-interpretation. Even the title appears rather vague and can be misleading.

Reviewers' comments:

Reviewer's Responses to Questions

**Comments to the Author**

1. If the authors have adequately addressed your comments raised in a previous round of review and you feel that this manuscript is now acceptable for publication, you may indicate that here to bypass the “Comments to the Author” section, enter your conflict of interest statement in the “Confidential to Editor” section, and submit your "Accept" recommendation.

Reviewer #2: All comments have been addressed

2. Is the manuscript technically sound, and do the data support the conclusions?

Reviewer #2: Partly

3. Has the statistical analysis been performed appropriately and rigorously? 

Reviewer #2: Yes

4. Have the authors made all data underlying the findings in their manuscript fully available?

Reviewer #2: Yes

5. Is the manuscript presented in an intelligible fashion and written in standard English?

Reviewer #2: Yes

6. Review Comments to the Author

Reviewer #2: The authors have addressed all the comments in detail and that has improved the manuscript substantially. I agree with the assessment of rare variants in CFI in European population, I am not convinced that this manuscript offers any new insights into the other population as authors claim. Authors have changed the title from “How does ethnicity affect prevalence of AMD-associated CFI rare genetic variants, and why does it matter?” to “An assessment of prevalence of Type 1 CFI rare variants in AMD, and why lack of genetic data from non-European ethnicities hinders development of new treatments and healthcare access.” However, the manuscript still does not offer much support for the non-European populations. It is well established that genetic studies from diverse ethnicity can offer new insights. Thus, in the lack of actual data of rare CFI variants from other populations in this manuscript, adding any statement on other populations especially in the title and conclusion seem to be overreaching, and authors should consider just focusing on the European population for which they have the data. The importance of other populations can be still included in the discussion, but I feel in its current form it could be misleading.

7. PLOS authors have the option to publish the peer review history of their article (what does this mean?). If published, this will include your full peer review and any attached files.

Reviewer #2: No

---

## [Author Response · Author response to Decision Letter 1]

13 Jul 2022

Rebuttal letter

Manuscript number: PONE-D-22-04791

Date: 13th July 2022

We thank the editor and the reviewer for their follow up comments on our manuscript to better improve communicating its intentions and limitations. Below is our response to each point raised by the academic editor and reviewers. We hope that we have now addressed the comments raised and that the manuscript will be now suited for publication.

Sincerely, 

On behalf of all authors,

Dr Amy V. Jones

…………………………………………………………………………………………………………………………………………………

Rebuttal to second round of Editor and Reviewer’s comments:

Additional Editor Comments:

In response to the Reviewer 2's continued concern about the lack of non-European populations in the manuscript, the authors should modify the text appropriately to avoid any over-reach or over-interpretation. Even the title appears rather vague and can be misleading.

Reviewers' comments:

Reviewer's Responses to Questions

Comments to the Author

1. If the authors have adequately addressed your comments raised in a previous round of review and you feel that this manuscript is now acceptable for publication, you may indicate that here to bypass the “Comments to the Author” section, enter your conflict of interest statement in the “Confidential to Editor” section, and submit your "Accept" recommendation.

Reviewer #2: All comments have been addressed

2. Is the manuscript technically sound, and do the data support the conclusions?

Reviewer #2: Partly

3. Has the statistical analysis been performed appropriately and rigorously? 

Reviewer #2: Yes

4. Have the authors made all data underlying the findings in their manuscript fully available?

Reviewer #2: Yes

5. Is the manuscript presented in an intelligible fashion and written in standard English?

Reviewer #2: Yes

6. Review Comments to the Author

Reviewer #2: The authors have addressed all the comments in detail and that has improved the manuscript substantially. I agree with the assessment of rare variants in CFI in European population, I am not convinced that this manuscript offers any new insights into the other population as authors claim. Authors have changed the title from “How does ethnicity affect prevalence of AMD-associated CFI rare genetic variants, and why does it matter?” to “An assessment of prevalence of Type 1 CFI rare variants in AMD, and why lack of genetic data from non-European ethnicities hinders development of new treatments and healthcare access.” However, the manuscript still does not offer much support for the non-European populations. It is well established that genetic studies from diverse ethnicity can offer new insights. Thus, in the lack of actual data of rare CFI variants from other populations in this manuscript, adding any statement on other populations especially in the title and conclusion seem to be overreaching, and authors should consider just focusing on the European population for which they have the data. The importance of other populations can be still included in the discussion, but I feel in its current form it could be misleading.

Author response:

We thank Reviewer 2 for further evaluation and recommendations to improve the main intention of this manuscript and welcome the opportunity to revise it further to better explain its limitations. 

Firstly, we have amended the title to not make mention of non-European populations, and it now reads; ‘An assessment of prevalence of Type 1 CFI rare variants in European AMD, and why lack of broader genetic data hinders development of new treatments and healthcare access.’

We have also updated the Abstract conclusion sentence to better reflect the data presented within the manuscript and highlight our conclusion that the gap in genetic data from other populations limits clinical development of FI targeted therapies, which we expand on more fully in the discussion section of the manuscript:

‘Conclusions: The relationship between Type 1 CFI rare variants increasing odds of AAMD are well established in Europeans, however the lack of broader genetic data in AAMD has adverse implications for clinical development and future commercialisation strategies of targeted FI therapies in AAMD’ (lines 78-80). 

7. PLOS authors have the option to publish the peer review history of their article (what does this mean?). If published, this will include your full peer review and any attached files.

Do you want your identity to be public for this peer review? For information about this choice, including consent withdrawal, please see our Privacy Policy.

Reviewer #2: No

 

…………………………………………………………………………………………………………………………………………………

Rebuttal to first round of Editor and Reviewer’s comments:

Editorial comments:

Journal Requirements:

Based on the PLOS ONE’s style requirements, we have fixed the section header formatting, bolding of supplemental table citation format and removed the abbreviations in the co-authors affiliations.

2. For studies reporting research involving human participants, PLOS ONE requires authors to confirm that this specific study was reviewed and approved by an institutional review board (ethics committee) before the study began. Please provide the specific name of the ethics committee/IRB that approved your study, or explain why you did not seek approval in this case.

Author response:

The Methods Ethics Statement has now been updated to provide the ethics committees that reviewed and approved the SCOPE and Finnish studies, which represent the novel datasets in this manuscript. Additionally, we also describe summary results from previously published studies, and now include reference to those publications that detail the ethics review committees/IRBs that reviewed and approved previously published datasets (lines 269 to 290). 

3. Thank you for stating the following in the Competing Interests: 

(The authors have declared that no competing interests exist:

Tom Southerington

Enni Makkonen

Roosa E. Kallionpää

Pauli Wihuri

Theresa Knopp 

Marco Hautalahti

Johanna Mäkelä

Arto Mannermaa

Erna Mäkinen

Anne-Mari Moilanen

Authors with competing interests

I have read the journal's policy and the authors of this manuscript have the following competing interests: 

Amy Jones: employee and share holder of Gyroscope Therapeutics Ltd

Darin Curtiss: employee and share holder of Gyroscope Therapeutics Ltd

Claire Harris: employee and share holder of Gyroscope Therapeutics Ltd, Research grant from RA Phaemaceutics (payment to institution), Royalty income from commercialized factor I ELISA; Hycult Biotech, consultancy income from Q32 Bio Inc, Chinook Therapeutics, and Biocryst Pharmaceuticals (all payment to institution), 

Nadia Waheed: employee and share holder of Gyroscope Therapeutics Ltd, grants from Carl Zeiss Meditec, Topcon, Regeneron, Heidelberg, Nidek, Optovue, consultancy income from Apellis, Nidek, Boehringer Ingelheim, stock in Ocudyne.)

We note that one or more of the authors have an affiliation to the commercial funders of this research study : Gyroscope Therapeutics Ltd, Research grant from RA Phaemaceutics (payment to institution), Hycult Biotech, Q32 Bio Inc, Chinook Therapeutics, Biocryst Pharmaceuticals

Author response:

We have revised the Funding Statement to state ‘Gyroscope Therapeutics Limited provided funding for the design and conduct of the SCOPE study, and also provided funding, approved the design, conduct, preparation, analysis and interpretation of the data presented in this manuscript, and decision to submit the manuscript for publication. This study was performed as part of the authors’ regular employment duties and the funder provided support in the form of salaries for authors (AJ, DC, CH, NW). (Lines 801-806).

Within your Competing Interests Statement, please confirm that this commercial affiliation does not alter your adherence to all PLOS ONE policies on sharing data and materials by including the following statement: ""This does not alter our adherence to PLOS ONE policies on sharing data and materials.” (as detailed online in our guide for authors http://journals.plos.org/plosone/s/competing-interests). If this adherence statement is not accurate and there are restrictions on sharing of data and/or materials, please state these. Please note that we cannot proceed with consideration of your article until this information has been declared.

Author response:

We have duly revised the cover letter to include a Funding Statement and Competing Interests Statement, and included text from the response to 2a above, and the sentence ‘The commercial interest described above does not alter our adherence to all PLOS ONE policies on data sharing and materials.’ (Lines 801-806).

Author response:

We included the Data Availability statement ‘Yes - all data are fully available without restriction,’ but this did not state we would provide this via a repository. All novel data presented are all contained within the manuscript and/or Supporting Information files already provided, and to this end we provided the statement ‘All relevant data are within the manuscript and its Supporting Information files.’ 

Therefore we understand that there are no changes to the existing Data Availability statement. Please advise if there needs to be further clarification. 

4. One of the noted authors is a group or consortium (Scope Study Group). In addition to naming the author group, please list the individual authors and affiliations within this group in the acknowledgments section of your manuscript. Please also indicate clearly a lead author for this group along with a contact email address.

Author response:

A lead author for the SCOPE study is now included in the authorship list, and a contact email address is provided in the Supportive Information section (line 797). 

Additional Editor Comments:

The two reviewers have raised important points of concerns and clarification. The key issue is that despite the title of effect of "ethnicity" no non-European cohort was included. In addition, the impact of these studies requires better clarification.

Reviewers' comments:

Reviewer's Responses to Questions

Comments to the Author

1. Is the manuscript technically sound, and do the data support the conclusions?

Reviewer #1: Yes

Reviewer #2: Partly

2. Has the statistical analysis been performed appropriately and rigorously?

Reviewer #1: Yes

Reviewer #2: Yes

3. Have the authors made all data underlying the findings in their manuscript fully available?

Reviewer #1: Yes

Reviewer #2: Yes

4. Is the manuscript presented in an intelligible fashion and written in standard English?

Reviewer #1: Yes

Reviewer #2: Yes

5. Review Comments to the Author

Reviewer #1: The paper focuses on a list of 18 rare variants in CFI associated with AMD, and investigating their frequencies in several populations, using genetic data sets from different organizations and countries.

The lack of genetic diversity, and the implication of this lack of information is a topic of discussion and investigation by many in the field of human genetics. At this point, it is a well-established fact that lack of proper genetic controls and background distribution hinders our understanding of human disease development, as well as development of treatment.

I welcome and agree with the sentiments of the authors on the immense need of a more genetically diverse data bases to be able accurately describe the genetic variability as well as disease linked variants in a given population (in this case rare variants contributing to AMD). With that being said, I feel that the authors focus most of the paper on the well described and curated European populations. The title of the paper is "How does ethnicity affect prevalence of AMD-associated CFI rare genetic variants, and why does it matter?" but throughout the paper the authors talk about a select list of variants that was selected based on European population and the other ethnicities are summarized in to we don't have enough data to show an association. I think that the calculations about the needed numbers of affected are a good starting point, but this will not be sufficient until the list of variants to be investigated is also matched to the population in question. For example, they mention the reported p.V412M variant in CFI identified in 3 Tunisian Jewish families, this is a different population from the Ashkenazi Jewish mentioned in the paper, so it should be investigated separately (or compared to other Northern African diaspora populations), so it is strange to me that it was noted together with the Ashkenazi analysis. The same paper also reported a CFI variant found in an Ashkenazi family (p.K441R) but this was not included in the analysis. The prevalent variants in each population may be very different so accounting for variants found in population A to investigate association in population B will always be an incomplete analysis.

I think that the part dedicated to European analysis should be significantly shorter, the contribution of CFI to AMD is not new to the field, and there is no major new findings there from my perspective. If more ethnicities can be added to the mix, I think this will be more informative, and can be used to guide organizations like the International Age-related Macular Degeneration Genomics Consortium (IAMDGC) and EYE-RISK in research design and focus.

There is also a place to mention that some of the variants described in the paper are not rare in all population, which should also be taken into account when we discuss the causality or pathogenicity of these variants. This is not necessarily within the scope of this paper, but I do think it should be mentioned.

Author response:

We thank reviewer 1 for their comments that this manuscript highlights an important gap in the genomics data landscape currently, that the lack of non-European genetic datasets hampers our global understanding around the contribution of rare variants to common diseases, here we discuss AMD as a good example. Due to the absence of genetic data from individuals with AMD from non-European background, we were unable to answer the question about how rare variation in the CFI gene affects AMD in global populations. The aim of the paper was to provide prevalence estimates for CFI rare variants associated with AMD in non-overlapping European AMD datasets, highlight the disparity that this assessment cannot be done in non-European datasets, and discuss why this is important for healthcare equity and developing new treatments for this devastating disorder. 

To better explain the intention of the manuscript, we updated the manuscript title to better signal the CFI rare variants were definable only in European populations; ‘An assessment of prevalence of Type 1 CFI rare variants in AMD, and why lack of genetic data from non-European ethnicities hinders development of new treatments and healthcare access’. 

The main novelty is the description of the frequency prevalences of rare CFI variant genotypes associated with low serum FI levels (referred to as ‘Type 1’), in two newly described AMD cohorts derived from the SCOPE natural history study, and a Finnish Biobank study (FINBB). Combined with prevalences derived from previously published AMD datasets, some of which required secondary analyses to determine (e.g. UK Biobank), this manuscript presents a comprehensive overview of the incidence of Type 1 rare CFI variants in European AMD. Understanding the size of this AMD sub-population has direct implications for the development of FI targeted gene therapies, and identifying target patient populations globally. 

In the results section, we present collective prevalences for five different European AMD datasets, two of which are novel (SCOPE and FINBB). There are currently no published comparable genetic datasets in any non-European ethnicity as far as we are aware from extensive searches on Pubmed. Therefore to answer the reviewers question, deriving comparable prevalence estimates in Ashkenazi Jewish, or Tunisian Jewish sporadic AMD populations, is not currently possible. We cite in the discussion that there have been a small number of publications describing sequencing studies in familial AMD from both these ethnic groups that have isolated rare CFI variants in affected, related individuals. These studies were restricted to investigating only a small number of families, and do not provide a full picture of the breadth of CFI rare variation that may exist in individuals with sporadic AMD. Later in the manuscript, we predict the sample sizes that might be required for screening to determine prevalence of CFI rare variants in AMD arising from the closest ethnicities to these groups, using background (control) datasets from gnomAD. As the reviewer explains, this could serve as a guide for organisations like IAMDGC and EYE-RISK for future research design and focus. 

Lastly, reviewer 1 states that there is ‘a place to mention that some of the variants described in the paper are not rare in all population’. The 18 CFI rare variant genotypes (as listed in S2 and S3 Tables) all individually have minor allele frequencies equal or less than 1% in background (European) populations, including non-Finnish Europeans from gnomAD. We believe this terminology using ‘rare’ in this context to be broadly correct as it aligns with the current definitions that rare variants are alternative forms of a gene that are present with a minor allele frequency (MAF) of less than 1%. However we acknowledge that the manuscript can be improved by better qualifying how we are using the term ‘rare’, therefore we added the following statement into the Methods section:

‘The term ‘rare’ refers throughout to variants that are equal or less than 1% minor allele frequency (MAF) in European background (control) populations (Cirulli and Goldstein Nat Rev Genet. 2010 Jun;11(6):415-25. doi: 10.1038/nrg2779).’ Lines 180-182. 

Reviewer #2: This manuscript focuses on understanding the role of rare variants in CFI on Age-related Macular Degeneration across different ethnicities (AMD). Both common and rare variants in CFI, which is a complete gene, have been associated with the risk of AMD. The authors focus on the rare variants that have been shown in earlier studies to affect the levels of CFI in serum and analyze them across multiple datasets. They find enrichment of CFI variants across all European cohorts and estimate the sample size needed to access the rare variants across different ethnicities. This reviewer has the following comets that I would like the authors to address:

1.The authors provide very little background on the 18 Types 1 variants analyzed in this study. It will be good to provide brief information on how many samples were analyzed, and what kind of sequencing was performed (exome or targeted sequencing) and what proportion of patients had rare variants in CFI.

Author response:

The patient numbers for those carrying rare CFI variant genotypes identified as Type 1 in the Java et al. (2020) paper are not representative of the overall population frequency because this study only cites results from in vitro testing of patient samples, which were selected on the basis of serum blood sample availability. Instead, we cited an earlier study (Kavanagh et al. (2015)) that describes CFI rare variant prevalences derived from the same underlying AMD patient dataset, and additionally from a matched control cohort (Table 1). The definition of Type 1 for the CFI rare variant genotypes analysed in vitro by Java et al. (2015) were on a limited number of samples. However the finding of low FI levels (‘haploinsufficiency’) in carrier blood serum accords with that from many previous observations reported in the literature, boosting confidence that these genotypes are functional. 

We have revised the text in the Methods and Discussion to explain these points more clearly, as follows:

Methods:

For CFI variant frequency analysis, we selected the 18 ‘Type 1’ CFI rare variant genotypes defined by Java et. al. (2020) (25), in a European AAMD cohort, on the basis 29 carrier patient serum blood samples underwent comprehensive serum-based in vitro functional assays measuring FI level and complement activity to determine ‘Type 1’ status. 

Because Java et. al. (2020) described the functional nature of different CFI rare variant from only in a minority of cases with serum available for functional evaluation of each genotype, more accurate variant frequency data was sourced from an earlier study examining the same underlying European AAMD (n=2,266) and control sample (n=1,400) datasets, as reported by Kavanagh et. al. (2015) (21), which employed a targeted next-generation sequencing strategy to capture the CFI coding and 5’ untranslated (UTR) and 3’ UTR regions (Table 1) (21). Lines 161-170.

Discussion:

The 18 Type 1 CFI rare variant genotypes investigated in this study were selected on the basis of comprehensive functional analyses showing their correlation with low serum FI levels in carrier blood serum (25). Due to the rare nature of these genotypes, and limited availability of CFI rare variant positive patient serum samples, only between 1 and 4 samples per genotype were tested in vitro. Functional characterisation using high-throughput serum-based functional assays like that described by Java et al. (2020) (25) are required to determine more definitively the clinical impact of rare genotypes identified in affected patients, where in silico analysis or prior clinical or functional characterisations are lacking. Despite the limited number of CFI rare variant positive samples used previously for functional characterisation, there is strong concordance in findings of FI haploinsufficiency driven by these genotypes in previous studies examining serum from other individuals carrying the nominated Type 1 CFI rare variant genotypes (23,24,55–58). Lines 514-524.

2.Based on Java et al., 2020 Type 1 variants were defined as those that demonstrated low FI antigenic levels and low iC3b generation, and Type 2 and 3 variants that exhibited normal levels but reduced function. Both of these categories can have an impact on the AMD disease outcome. Then why did the authors choose to follow only type-1 variants?

Author response:

Those CFI rare variant genotypes categorised as Type 1 are considered to have the most direct functional impact by reducing FI levels systemically, and carriers of these genotypes who are affected by AMD are hypothesised to be the sub-population of patients who would maximally benefit from FI-targeted gene therapy. We acknowledge that other CFI rare variant genotypes, categorised as Type 2 and 3, also impact negatively by reducing protein function but not levels. However, establishing a dysfunctional consequence from Type 2 and 3 genotypes on resulting FI protein requires specialised complement binding assays. This has made the functional consequence of Type 2 and 3 genotypes more challenging to characterise than Type 1 genotypes, where measuring FI serum levels is technically more straightforward. Because of this, there is a greater body of evidence in the literature supporting correlations between Type 1 genotypes and (low) FI level. The level of risk of AAMD conferred by Type 1 CFI rare variant genotypes has been more deeply characterised compared to Type 2 and 3 CFI rare variants, and (limited) data suggests Type 1 genotypes may confer a greater magnitude of disease risk, further confirming Type 1 genotypes as being more characterised in terms of their contribution to disease. 

To address this point raised by Reviewer 2 in the manuscript, we added the following paragraph to the discussion:

In addition to Type 1 CFI rare variants, Java et al. (2020) also described other CFI rare variant genotypes that produce FI proteins with FI levels in the normal range but had varying degrees of reduced enzymatic activity, determined using specialised in vitro complement binding assays (Ref). Termed ‘Type 2’ and ‘Type 3’ CFI rare variants, it is plausible that carriers of these genotypes may also benefit from targeted FI gene therapy to restore normal protein function to the retina, rebalancing the CS. However, there is low concordance or a lack of independent verification in the literature for which genotypes meet the ‘Type 2’ or ‘Type 3’ categorisation, and less is known about how these genotypes confer elevated risk of AAMD, compared to Type 1 genotypes. For example, the CFI rare variant p.Arg406His was categorised as a Type 3 indicating moderate protein dysfunction, but this is contentious in the literature (64,65). Additionally, p.Arg406His has been previously reported as protective for AAMD, decreasing risk for disease not increasing risk, like other genotypes (OR: 0.1 (22). Because of this, prevalences of CFI Type 2 and 3 genotypes were not investigated here, but carriers of Type 2 and 3 genotypes remain an important patient sub-group of AMD for the clinical development of FI-targeted therapies. Lines 536-549. 

3.While the UK biobank, SCOPE, and genomeAD database have sequencing data, IAMGDC data was based on the genotypes. They did include the rare variants, but it was not as comprehensive as sequencing-based discovery and is limited by the variants present in the array. This should be clearly stated.

Author response:

We have updated the discussion to better call out the differences in the sequencing strategies between the studies as suggested by Reviewer 2, by adding a new paragraph:

In addition, differences in genotyping strategy employed by each study may have affected prevalence estimates by the ability to detect all protein-altering variation at CFI; SCOPE, UK Biobank, FINBB and the Kavanagh et. al. (2015) dataset (21) all used a targeted sequencing approach, but the IAMDGC study (33) relied on a custom modified HumanCoreExome SNP array, with enriched content for protein-altering variants in AAMD risk genes identified in previous targeted next-generation sequencing (NGS) and whole genome sequencing (WGS) studies (41,47). Despite the less comprehensive genotyping strategy, the IAMDGC reported comparable collective variant frequencies of Type 1 CFI rare variants to the other studies, indicating that there was a good representation of the genotypes on the array. Lines 424-432. 

We have also updated S1 Table with an additional column to describe the methods for variant identification utilised in each AAMD study. 

4.The odds ratio for type-1 variants is high and closer to the original analysis Kavanagh et al., 2015. Could the authors offer an explanation for this finding? Could it be attributed to better phenotyping of the AMD and controls in IAMDGC and Kavanagh et al? I would like the authors to include this in the discussion.

Author response:

Since this manuscript was created we have performed a new datacut on the SCOPE study, and the sample size has increased to 3,243 patients with the odds ratio (OR) for AAMD increasing to 7.8. We believe this greater sample size improves the manuscript; therefore we have taken the opportunity to revise the SCOPE data presented in Table 1 and S2 Table with the updated frequencies, and we have updated the values cited throughout the manuscript. 

This update of the SCOPE study OR to 7.8 brings it closer to that reported by the IAMDGC and Kavanagh et. al. (2015) studies, who reported ORs of 4.2 and 11.4, respectively. We agree that the stronger ORs may be reflective of the more stringent phenotyping of AMD performed by retinal specialists, in comparison to the UK Biobank AMD study that reported a weaker OR of 3.1, which only relied on self-reported AMD status. We have updated the manuscript discussion to describe this:

The stronger ORs observed in the Kavanagh et. al. (2015) (21), SCOPE and FINBB studies may be reflective of the more stringent phenotyping undertaken by retinal specialists (for FINBB this was in a healthcare registry setting), to confirm each patient’s AAMD diagnosis and disease stage. This is in comparison to the UK Biobank AAMD study which relied only on an individual’s self-reported AMD status to define cases and controls, and reported a weaker OR of 3.1. Lines 417-422. 

5.The focus of the paper centers around finding the difference in the frequency of the rare variant based on ethnicity. This is well-documented that rare variants tend to be specific to the population. All of the data presented here are from the publicly available data which shows the frequency differences across populations, which is not surprising. Authors then predict the estimate of the sample size needed to study type I variants across different ethnicity. However, the motivation for this is not clear. Different ethnicities are likely to have their unique, rare variants. It is also likely that in some populations the disease may not be driven by the rare variants in CFI. Could the authors explain the rationale behind this?

Author response:

One of the key motivations for this study was exploring the prevalence of Type 1 CFI rare variants linked to FI protein haploinsufficiency, in non-European AMD datasets. Characterising this subgroup of AMD patients is important for clinical development and commercialisation of targeted FI therapies, as AMD patients carrying Type 1 CFI rare variants are hypothesised to be most likely to benefit. These analyses try to define the size and distribution of the (Type 1) target populations that might exist, outside of European AMD. But as there is a lack of genetic data in AMD from non-Europeans, we are unable currently to answer this question. To try to drive the field to address this, we present power calculations for possible sample sizes of AMD patients from different non-European backgrounds that would need to be screened prospectively to determine wither the Type 1 CFI rare variants previously characterised, are present or likely absent. 

Of course, this does not preclude there being other CFI rare variant genotypes that are unique to a given non-European AMD population, which may be associated with low serum FI levels and cause complement dysregulation. However these genotypes firstly need to be identified by new studies, then functionally characterised to prove a robust association with low FI levels, before they can be used to direct patient stratification to clinical trials testing FI-targeted therapies that are seeking to enrol genetically defined sub-populations. To better describe this, we added to the introduction and discussion:

Introduction:

This may stimulate the field to generate more sequencing studies in individuals with AAMD from diverse ethnic backgrounds, further supporting the clinical development of novel therapies which could reach a wider range of AAMD patients. Lines 151-153

Discussion:

For clinical development of FI-targeted therapies to reach global populations, there needs to be more sequencing studies conducted on AMD populations from diverse ethnicities and geographies, in order to identify population-specific CFI rare variants that can be functionally characterised for their effect on the FI protein, and CS pathway. Any new CFI rare variant genotypes identified could be added to the Type 1 (or Type 2 and 3) lists, and individuals with AAMD who carry these variants may ultimately be stratified into clinical trials testing FI targeted treatments that are seeking to enrol genetically defined sub-populations of AMD. This study describes power calculations used to estimate the likely sample sizes of individuals with AMD from different ethnic backgrounds, that may be required to detect (European) Type 1 CFI rare variants if present. However this does not address the sample sizes required to detect functional CFI rare variants that might be unique to a different ethnic group, as it is well documented that rare variation is more likely to be restricted to a given population (35,36). Furthermore, the sample size analyses performed here were limited to those genetic backgrounds defined by GnomAD, and there are more diverse ethnic groups that remain to be characterised. Lines 554-567. 

AAMD is a complex disorder, with multiple risk factors that include genetics but also non-genetic factors, such as age, environment, diet and sunlight exposure. We have updated the Introduction to better describe the interplay between genetics and environment:

AMD is a complex disease with a multifactorial aetiology influenced by age and environmental factors such as diet and sunlight exposure, and a positive family history (3,5). Lines 93-95. 

6. PLOS authors have the option to publish the peer review history of their article (what does this mean?). If published, this will include your full peer review and any attached files.

Do you want your identity to be public for this peer review? For information about this choice, including consent withdrawal, please see our Privacy Policy.

Reviewer #1: No

Reviewer #2: No

---

## [Editor Report · Decision Letter 2]

15 Jul 2022

An assessment of prevalence of Type 1 CFI rare variants in European AMD, and why lack of broader genetic data hinders development of new treatments and healthcare access

PONE-D-22-04791R2

Dear Dr. Waheed,

We’re pleased to inform you that your manuscript has been judged scientifically suitable for publication and will be formally accepted for publication once it meets all outstanding technical requirements.

Kind regards,

Anand Swaroop

Academic Editor

PLOS ONE
---

## [Editor Report · Acceptance letter]

26 Aug 2022

PONE-D-22-04791R2 

An assessment of prevalence of Type 1 *CFI* rare variants in European AMD, and why lack of broader genetic data hinders development of new treatments and healthcare access 

Dear Dr. Waheed:

I'm pleased to inform you that your manuscript has been deemed suitable for publication in PLOS ONE. Congratulations! Your manuscript is now with our production department. 

Kind regards, 

on behalf of

Dr. Anand Swaroop 

Academic Editor

PLOS ONE